

# Disorder in AdS$_3$/CFT$_2$

Moritz Dorband[1,2★], Daniel Grumiller[3,4], René Meyer[1,2] and Suting Zhao[1,2]

**1** Institute for Theoretical Physics and Astrophysics,
Julius–Maximilians–Universität Würzburg,
Am Hubland, 97074 Würzburg, Germany
**2** Würzburg-Dresden Cluster of Excellence ct.qmat
**3** Institute for Theoretical Physics, TU Wien,
Wiedner Hauptstrasse 8-10/136, A-1040 Vienna, Austria
**4** Theoretical Sciences Visiting Program, Okinawa Institute of Science and Technology,
Graduate University, Onna, 904-0495, Japan

★ moritz.dorband@physik.uni-wuerzburg.de

## Abstract

We perturbatively study marginally relevant quenched disorder in AdS$_3$/CFT$_2$ to second order in the disorder strength. Using the Chern–Simons formulation of AdS$_3$ gravity for the Poincaré patch, we introduce disorder via the chemical potentials. We discuss the bulk and boundary properties resulting from the disorder-averaged metric. The disorder generates a small mass and angular momentum. In the bulk and the boundary, we find unphysical features due to the disorder average. Motivated by these features, we propose a Poincaré–Lindstedt-inspired resummation method. We discuss how this method enables us to remove all of the unphysical features and compare with other approaches to averaging.

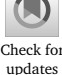

# 1  Introduction

Strongly interacting quantum systems are of interest in both high-energy physics and condensed matter physics. While in the description of condensed matter systems idealisations such as the perfect crystal are often useful to understand aspects of the physics of the electron system, taking into account the effect of impurities and disorder implemented by lattice defects is important to fully describe the physics as measured in experiments. In particular, the presence of random disorder completely changes the low energy transport properties of otherwise free quantum particles, a phenomenon nowadays known as Anderson localisation [1]. Anderson localisation states that in disordered lower dimensional systems, disorder leads to a complete termination of diffusive processes, making a metal a perfect insulator in the thermodynamic limit. While this was shown for non-interacting systems in [1], it is an open question how interactions will affect the localisation transition, which is a contemporary open problem in theoretical physics known as many–body localisation (for reviews see [2,3]).

While interactions can be included perturbatively into an already disordered system, an interesting question is whether many–body localisation persists in the non-perturbative regime, in particular, if the interacting but non-disordered phase would be gapless. A promising approach to describe strongly interacting systems, and hence to answer this question, is provided by the AdS/CFT correspondence (a.k.a. gauge/gravity duality or simply holography) [4]. The AdS/CFT correspondence relates strongly interacting quantum field theories (QFTs) to weakly interacting gravitational theories in asymptotically Anti–de Sitter (AdS) spacetimes. In the limit of large central charges, the generating functional of the dual strongly interacting quantum field theory is defined on the asymptotic boundary of the AdS spacetime via the semiclassical gravitational path integral with fixed boundary conditions [5,6]. In particular, each field in the bulk encodes both the source and response (expectation value) of an operator in the dual boundary QFT. In this work, we will disorder the boundary sources by drawing them randomly from a Gaussian ensemble, solve the gravity equations of motion for the bulk spacetime in each disorder realisation, and analyse the resulting disorder-averaged bulk geometry. We work in the realm of the $AdS_3/CFT_2$ correspondence, and in particular use the $SL(2,\mathbb{R}) \times SL(2,\mathbb{R})$ Chern–Simons formulation [7,8] of three-dimensional Einstein–Hilbert gravity with a negative cosmological constant.

The possibility of introducing quenched disorder in AdS/CFT was first adumbrated in [9] (see [10] for a review). Momentum–relaxing impurities have been introduced in AdS/CFT even earlier in [11], and the replica trick has first been used in [12] to calculate the effect of disorder in AdS/CFT. Refs. [13–16] studied a disordered scalar in AdS spacetime. In [13] the effects of disorder on a holographic superconductor were studied by introducing a random chemical potential on the boundary. In [14], a scalar field coupled to three-dimensional AdS

gravity was considered, where the source dual to the scalar operator was disordered. The scalar source was disordered along the spatial boundary dimension, and several different ensemble realisations for the disorder were discussed in [14]. Disorder was finally introduced in [13, 14] by an ansatz using superimposed plane waves with equally distributed random phases. By solving Einstein's equations to second order in the disorder strength, the backreaction of the scalar field on the pure AdS background was determined. The resulting metric averaged over disorder realisations showed a logarithmically divergent behaviour in the infrared (IR) region in the bulk. The logarithmic behaviour can be traced back to the disorder being marginally relevant in the sense of Harris' criterion [17]. The Poincaré–Lindstedt resummation method was then employed to remove this divergence. The resummed metric admitted an IR Lifshitz scaling fixed point, with the dynamical exponent departing from the relativistic value quadratically in the disorder strength. The quartic correction in the disorder strength was also calculated analytically. The analytic results were confirmed by a numerical analysis for non-perturbative disorder strength. In [15], this analysis was extended to finite temperature, i.e., to a planar AdS black hole background, both in $D = 3$ and $D = 4$. A scaling of the low–temperature entropy density with temperature, with the exponent fixed by the Lifshitz scaling exponent found at zero temperature [14], was constructed both analytically and numerically. Furthermore, it was realised [18, 19] in a non-perturbative treatment of the model of [14] that the Lifshitz scaling is only approximate, and non-perturbative effects lead to a different IR fixed point independent of the disorder strength at exponentially small energy scales. In [16], extremal planar AdS black holes, and in particular their near horizon behaviour, in $D = 4$ Einstein–Maxwell theory were studied. Furthermore, disorder in a p-wave holographic superconductor has been investigated in [20], and a disorder-induced holographic metal-insulator transition has been found in [21].

Einstein–Maxwell theory in $D = 4$ with a disordered boundary chemical potential for the U(1) charge density was investigated in [22]. AdS gravity supplemented with a U(1) gauge field in the bulk describes a boundary state at finite U(1) charge density at the boundary condition, with the chemical potential being the constant mode in the time component of the bulk chemical potential. In AdS/CFT, the full U(1) gauge field in the bulk is holographically dual to a global conserved Noether current in the boundary QFT. By computing the backreaction of the disordered vector potential on the metric to second order in the disorder strength, the effect of disorder on the bulk spacetime geometry was determined analogously as in [14]. From this metric, a positive second-order correction to the conductivity of the boundary theory was found. This result is consistent with [23], where a lower bound for the boundary conductivity of disordered Einstein–Maxwell AdS gravity was derived, implying that any source of disorder will raise the average value of the boundary conductivity in Einstein–Maxwell AdS gravity. Similarly, a lower bound on the thermal conductivity in Einstein–Maxwell–Dilaton theory was found in [24], provided the dilaton potential is bounded from below. Preceding [23], it was shown in [25] that random disorder in Einstein–Maxwell–Dilaton theory has a strong effect on the DC resistivity. Both DC and AC conductivities in $D = 4$ have also been studied in a holographic lattice setup in Einstein–Maxwell [26] and Einstein–Maxwell–Dilaton theory [27]. More recently, the effect of disorder on phase transitions has been discussed in [28, 29]. [28] showed that the presence of disorder in $d > 4$ Euclidean CFTs leads to new fixed points of the renormalization group which are in general not scale-invariant. Furthermore, in [29] it was shown that including disorder into a model for a holographic Weyl semimetal leads to a smearing of the quantum phase transition between the topologically trivial and non-trivial phases.

The possibility of Anderson localisation in holographic systems has been discussed in [23, 30–32]. In [23] it was argued that for a certain class of holographically mean-field disordered systems based on Einstein-Maxwell theory, there is no disorder-driven metal-insulator phase

transition. On the contrary, it was found in [30, 31] that adding couplings between the translation breaking sector and the electric charge sector indeed does yield disorder-driven metal-insulator transitions. Furthermore, by analysing a scalar field on a disordered background metric, [32] found that the two-point function receives corrections dependent on the disorder strength which, for particular implementations of the disorder, lead to large long-range correlations suggesting that the system flows to a new disorder-driven fixed point.

A related question is whether the notion of Anderson localisation, originally derived in quantum mechanical systems of free or weakly interacting particles, can in principle be analysed within a holographic setup. A key question is whether the quantum nature of the quantum interference of single-particle wave functions leading to the Anderson insulator is crucial for the phenomenon, or whether classical wave interference, as should occur in the bulk of a disordered AdS spacetime before averaging, is sufficient. Disorder induced localisation also appears in systems of electromagnetic [33–35] and acoustic waves [36–38]. From these studies we conclude that Anderson localisation requires interference, but not necessarily quantum interference, and that holographic theories should be useful for studying Anderson localisation.

In this work, we use the $AdS_3/CFT_2$ correspondence to study holographically disordered strongly interacting two-dimensional conformal field theories (2*D* CFTs). We employ the first–order formulation of three-dimensional Einstein gravity with a negative cosmological constant as $SL(2, \mathbb{R}) \times SL(2, \mathbb{R})$ Chern–Simons theory [7, 8]. Choosing asymptotic Brown–Hennaux boundary conditions [39], the dual QFT is a 2*D* CFT at large central charge, with symmetry given by two copies of the Virasoro algebra [39]. As described in more detail below, we introduce quenched static disorder by disordering the sources encoded in the asymptotic behaviour of the bulk Chern–Simons gauge fields, i.e., the chemical potentials coupling to the left- and right-moving Chern–Simons currents. A crucial advantage of this approach is the exact solvability of the bulk Chern–Simons theory [40], providing an explicit expression for the bulk solution for each disorder realisation, and hence allowing us to derive our results analytically to second order in the disorder strength.

Having introduced disorder into the system, we calculate the disorder-averaged bulk metric to quadratic order in the disorder strength. We then analyse both bulk and boundary properties of the resulting geometry. We find that there are curvature singularities in the IR region. As we will discuss in more detail in section 2.2, these singularities are of a "bad" type and are therefore deemed unphysical. The presence of the singularity implies a breakdown of the perturbation expansion in the disorder strength. Furthermore, our averaged metric does not satisfy the $AdS_3$ vacuum Einstein equations any longer for non-vanishing disorder strength. The averaged metric is moreover found to contain closed timelike curves. In the region containing closed timelike curves, a breakdown of the semiclassical approximation for a probe string is found, one of the three criteria for a "bad singularity" mentioned in [41]. In the field theory dual, we find that the energy-momentum tensor has a trace anomaly, which however cannot be related to the Ricci curvature of the boundary metric via the usual trace anomaly equation. Finally, depending on the relative strengths of the disorder parameters for left and right moving sectors as well as on the length of the entangling interval, we find that the quantum null energy condition (QNEC), as well as the null energy condition (NEC), can be violated.

To remedy all these issues, we perform a resummation inspired by (but different from) the Poincaré–Lindstedt technique used in [14, 16]. With this method, we restore finite curvature throughout the entire bulk. The resulting resummed averaged metric satisfies the $AdS_3$ vacuum Einstein equations, up to second order in the disorder strength. The resummed metric furthermore no longer contains closed timelike curves and yields a well-defined semiclassical approximation for a string probing the interior of the averaged spacetime. The dual theory's

energy-momentum tensor is now traceless, following the usual relation between the boundary Ricci curvature and the trace anomaly. Also, the QNEC condition for the resummed metric is not only satisfied but saturated. This single resummation ansatz solving all of these problems at once suggests this is the correct bulk procedure to obtain a reasonable holographic disorder–averaged state.

Having discussed the disorder-averaged metric and its resummation in great detail, we also study performing the average directly in the Chern–Simons theory. The resulting averaged connections fail to satisfy the gauge flatness conditions. Calculating the metric for the averaged connections, we again find a curvature singularity in the IR, albeit of different behaviour than for averaging the metric. Moreover, again we find that the energy-momentum tensor of the dual description has a trace anomaly which does not satisfy the usual relation to the boundary Ricci curvature, i.e. the trace anomaly equation. As for the averaged metric, we perform a resummation of the averaged connections. By a minimal modification of the lowest weight components of the $t$-components of the connection, we can restore all of the aforementioned properties: the resummed connections are gauge flat, the corresponding metric is not divergent and the dual state satisfies the trace anomaly equation, in particular with both sides of the equation vanishing. In general, averaging and calculating the metric from the connections does not commute. Remarkably, the resummed metric obtained before and the metric following from the resummed connections are, up to an overall sign in the $t - \phi$ component, identical. The sign difference only appears for the case of different left- and right-moving disorder strengths $\epsilon \neq \bar{\epsilon}$, and may be removed by a time reversal transformation $t \to -t$ when passing between the two formalisms. Since even for non-rotating and hence parity and time reversal invariant backgrounds, $\epsilon \neq \bar{\epsilon}$ induces rotation and breaks time reversal and parity explicitly, it does not come as a surprise that a time reversal non-trivially transforms disorder-averaged backgrounds into each other. This is yet another instance of averaging and disordering not commuting, this time between the Chern–Simons and metric formalisms, which are equivalent to each other within each disorder realisation before averaging, but are found to differ by a sign after averaging.

Finally, we compare our results with those obtained by first calculating observables for each realisation of the disorder and then averaging only the final result. In particular, we obtain an averaged energy-momentum tensor that differs from the one generated through the Poincaré–Lindstedt-inspired method. The main disadvantage of this, otherwise straightforward, alternative is that there is no way to associate an effective metric with the resulting energy-momentum tensor. In particular, the averaged boundary metric does not provide a source for the averaged boundary energy-momentum tensor.

This paper is organised as follows: in section 2 we briefly review the Chern–Simons formulation of three-dimensional gravity in AdS and then introduce the disorder implemented in section 2.1. We discuss our results from the bulk perspective in section 2.2 and from the boundary perspective in section 3. Afterwards, we present a Poincaré–Lindstedt inspired resummation of these results in section 4. In section 5 we first analyse the average performed directly on the level of the connections of the Chern–Simons theory. After discussing the results of this averaging procedure, we also present a resummation in the Chern–Simons formulation. In the second part of this section, we compare our results with those obtained by first calculating observables for each realisation of the disorder and then averaging only the final result. We conclude and give an outlook towards future work in section 6.

## 2 Disorder in SL(2,R) Chern–Simons gravity

Classical Einstein–Hilbert gravity with negative cosmological constant in three spacetime dimensions can equivalently be formulated as a Chern–Simons gauge theory [7, 8]. We use this formalism to conveniently introduce disorder into the system, while also ensuring asymptotic AdS behaviour. Before discussing the disorder setup, we will therefore briefly review the Chern–Simons formulation of $AdS_3$ gravity.

The three-dimensional Einstein–Hilbert action on the classical level can equivalently be expressed as the difference of two Chern–Simons actions [7, 8]

$$S_{\text{EH}}[A,\bar{A}] = \frac{k}{4\pi}\int_{\mathcal{M}} \text{tr}\left(A\wedge dA + \frac{2}{3}A\wedge A\wedge A - \bar{A}\wedge d\bar{A} - \frac{2}{3}\bar{A}\wedge\bar{A}\wedge\bar{A}\right), \tag{1}$$

where the Chern–Simons level[1] $k = \frac{1}{4G} = \frac{c}{6}$ is related to Newton's constant $G$ and the Brown–Henneaux central charge $c$. The gauge fields $A$ and $\bar{A}$ are linear combinations of the dreibein $e^a{}_\mu$ and the dualised spin connection $\omega^a{}_\mu = \frac{1}{2}\epsilon^{abc}\omega_{bc\mu}$. Furthermore, the gauge fields are charged under the isometry group of the spacetime under consideration. In this paper, we consider AdS space which in $D = 3$ has the isometry group $SO(2,2) \simeq SL(2,\mathbb{R}) \times SL(2,\mathbb{R})$. Explicitly, the gauge fields are realised as

$$A = A^a T_a = (\omega^a + e^a)T_a, \quad \text{and} \quad \bar{A} = \bar{A}^a T_a = (\omega^a - e^a)T_a, \tag{2}$$

where $T_a$ are generators of $\mathfrak{sl}(2,\mathbb{R})$, satisfying $[T_a, T_b] = (a-b)T_{a+b}$. For explicit calculations, we work with the fundamental representation for $\mathfrak{sl}(2,\mathbb{R})$ (see appendix A).

We consider manifolds whose topology is a solid cylinder, with radial coordinate $\rho \in \mathbb{R}$, time $t \in \mathbb{R}$ and angular coordinate $\phi \in [0, 2\pi]$. To ensure asymptotic AdS behaviour, we impose Brown–Henneaux boundary conditions on the gauge fields as described in [40] (see also [42, 43] for reviews). To do so we first write the fields in radial gauge

$$A = b^{-1}(d+a)b, \qquad \bar{A} = b(d+\bar{a})b^{-1}. \tag{3}$$

The functions $a$ and $\bar{a}$ depend only on the boundary coordinates and capture all of the state dependence. The radial dependence is captured by the group element $b$ only, which we choose as

$$b = \exp(\rho T_0). \tag{4}$$

Brown–Henneaux boundary conditions are then achieved by imposing the following form on the state-dependent functions $a$ and $\bar{a}$

$$a_t = \mu T_+ - \partial_\phi \mu T_0 + \left(-\frac{2\pi}{k}\mu\mathcal{L} + \frac{1}{2}\partial_\phi^2\mu\right)T_-, \tag{5}$$

$$a_\phi = T_+ - \frac{2\pi}{k}\mathcal{L}T_-, \tag{6}$$

$$\bar{a}_t = \left(\frac{2\pi}{k}\bar{\mathcal{L}}\bar{\mu} + \frac{1}{2}\partial_\phi^2\bar{\mu}\right)T_+ + \partial_\phi\bar{\mu}T_0 + \bar{\mu}T_-, \tag{7}$$

$$\bar{a}_\phi = \frac{2\pi}{k}\bar{\mathcal{L}}T_+ + T_-, \tag{8}$$

where the gauge-flatness conditions on $A$ and $\bar{A}$ following from (1),

$$dA + A\wedge A = 0 = d\bar{A} + \bar{A}\wedge\bar{A}, \tag{9}$$

---

[1]We set the AdS radius to one.

have been used. Both barred and non-barred sector contain two free functions each, the chemical potentials $\mu, \bar{\mu}$ and their corresponding conserved currents $\mathcal{L}, \bar{\mathcal{L}}$, respectively. As shown in [40], a well-defined variational principle only allows the latter ones to vary and generate the Virasoro algebras for the barred and non-barred sectors.

By the flatness conditions (9), the conserved currents are determined completely by their associated chemical potentials

$$\partial_t \mathcal{L} = \mu \partial_\phi \mathcal{L} + 2\mathcal{L} \partial_\phi \mu - \frac{k}{4\pi} \partial_\phi^3 \mu \,, \tag{10}$$

$$\partial_t \bar{\mathcal{L}} = \bar{\mu} \partial_\phi \bar{\mathcal{L}} + 2\bar{\mathcal{L}} \partial_\phi \bar{\mu} + \frac{k}{4\pi} \partial_\phi^3 \bar{\mu} \,. \tag{11}$$

For the vanilla choice of chemical potentials, $\mu = 1 = -\bar{\mu}$, these conservation equations are the usual holographic Ward identities $\partial_- \mathcal{L} = 0 = \partial_+ \bar{\mathcal{L}}$, with $x^\pm = t \pm \phi$.

In this work, to solve these conservation equations, we assume static chemical potentials, $\partial_t \mu = 0 = \partial_t \bar{\mu}$. Then the solutions for the currents are given by

$$\mathcal{L} = \frac{F\left[t + \int^\phi \frac{\mathrm{d}u}{\mu(u)}\right]}{\mu^2} + \frac{k}{4\pi} \left( \frac{\partial_\phi^2 \mu}{\mu} - \frac{1}{2} \frac{(\partial_\phi \mu)^2}{\mu^2} \right) \,, \tag{12}$$

$$\bar{\mathcal{L}} = \frac{G\left[t + \int^\phi \frac{\mathrm{d}v}{\bar{\mu}(v)}\right]}{\bar{\mu}^2} - \frac{k}{4\pi} \left( \frac{\partial_\phi^2 \bar{\mu}}{\bar{\mu}} - \frac{1}{2} \frac{(\partial_\phi \bar{\mu})^2}{\bar{\mu}^2} \right) \,. \tag{13}$$

In general, $F$ and $G$ are arbitrary functions depending on the indicated combination of $t$ and $\phi$.

From the above equations (12) and (13), the Poincaré AdS solution is obtained by the vanilla choice, $\mu = 1 = -\bar{\mu}$, and furthermore $F = 0 = G$, such that $\mathcal{L} = 0 = \bar{\mathcal{L}}$. Using the well-known relation between metric and gauge fields

$$g_{\mu\nu} = \frac{1}{2} \mathrm{tr}\left[ \left(A - \bar{A}\right)_\mu \left(A - \bar{A}\right)_\nu \right] \,, \tag{14}$$

we find the Poincaré patch metric

$$\mathrm{d}s^2 = -e^{2\rho} \mathrm{d}t^2 + \mathrm{d}\rho^2 + e^{2\rho} \mathrm{d}\phi^2 \,. \tag{15}$$

In the next subsection, we exploit the chemical potentials to introduce disorder onto the background given by (15). For the remainder of this paper, we use the holographic coordinate $\rho = -\ln z$.

## 2.1 Disorder setup

Our approach to introducing disorder into (15) is inspired by earlier work on disorder [14]. In our case, the chemical potentials $\mu$ and $\bar{\mu}$ serve as sources for disorder, which is considered analogously to [14]:

$$\mu = 1 + \epsilon f(\phi) = 1 + \frac{\epsilon}{\sqrt{N}} \sum_{n=1}^{N} \cos\left( \frac{n}{N} \phi + \gamma_n \right) \,, \tag{16}$$

$$\bar{\mu} = -1 + \bar{\epsilon} f(\phi) = -1 + \frac{\bar{\epsilon}}{\sqrt{N}} \sum_{n=1}^{N} \cos\left( \frac{n}{N} \phi + \gamma_n \right) \,. \tag{17}$$

The random phases $\gamma_n$ are equally distributed in the interval $[0, 2\pi)$. The chemical potentials are then periodic in $2\pi N$, only.[2]

Averaging an arbitrary function $H$ over these random phases is defined by integration over all random phases $\gamma_n$ and taking the large $N$ limit afterwards

$$\langle H(\{\gamma_n\})\rangle = \lim_{N \to \infty} \int_0^{2\pi} \prod_{n=1}^{N} \frac{d\gamma_n}{2\pi} H(\{\gamma_n\}). \tag{18}$$

The average, which we denote by $\langle \cdot \rangle$, is defined such that

$$\langle f(\phi) \rangle = 0, \tag{19}$$

leading to

$$\langle \mu \rangle = 1 = -\langle \bar{\mu} \rangle. \tag{20}$$

Therefore, to zeroth order in the disorder strengths $\epsilon$ and $\bar{\epsilon}$, this reproduces the vanilla choices $\mu = 1 = -\bar{\mu}$ made earlier for (15). More generally, it can be shown that every odd power of $f(\phi)$ averages to zero. Since the metric is bilinear in the gauge fields, and thereby also in the chemical potentials, we obtain terms with non-trivial averages by employing (14). Note that there certainly are also other ways to implement the averaging procedure. As is clear from the relation between the metric and the gauge fields (14), directly averaging the gauge field before computing the metric components is a distinct procedure. While the metric formulation and the Chern–Simons formulation of AdS$_3$ gravity are classically equivalent, the metric being bilinear in the gauge fields implies that this equivalence breaks during the averaging procedure.

The disorder that we introduced through random phases describes Gaussian noise. Computing for example the two-point function of $f$ results in

$$\langle f(\phi) f(\theta) \rangle = \frac{1}{2} \delta(\phi - \theta). \tag{21}$$

In principle, we are now equipped to calculate the average of the metric components from (14), using (16) and (17) to introduce disorder. In practice, this calculation involves integrals over inverse powers of $\mu$ and $\bar{\mu}$ (see (12), (13)), for which a closed expression could not be found. Therefore, since in the present work, we mostly aim for analytic results, we use $\epsilon$ and $\bar{\epsilon}$ as perturbative handles to expand the metric components to second order. This enables us to perform an analytic computation of the averages. While it is technically feasible to go beyond the quadratic order, we have checked that going to $\mathcal{O}(\epsilon^4)$ does not lead to a qualitative change of the results discussed in the following sections (see App. C) which is why we restrict our analytic analysis to the second order in the disorder strength.

While averaging the metric components to second order, there are four different combinations of $f$ and its derivatives that occur. The averages of these combinations are given by

$$\langle f^2 \rangle = \frac{1}{2}, \quad \langle (\partial_\phi f)^2 \rangle = \frac{1}{6}, \quad \langle f \partial_\phi^2 f \rangle = -\frac{1}{6}, \quad \langle (\partial_\phi^2 f)^2 \rangle = \frac{1}{10}. \tag{22}$$

In our work, we take the same random phases for $\mu$ and $\bar{\mu}$, i.e., we do not introduce random phases $\gamma_n$ and $\bar{\gamma}_n$, where we could allow for different intervals, say $[0, 2\pi)$ for $\gamma_n$ and $[\alpha, 2\pi + \alpha)$ for $\bar{\gamma}_n$. In that way, relative phases between $\epsilon$ and $\bar{\epsilon}$ can be included. However, at the level of our current analysis, this does not yield an immediate advantage. For more details on the effect of relative phases, we refer the reader to appendix B.

---

[2]As a consequence, the chemical potentials are not single-valued anymore, in line with the disorder ansatz by Hartnoll and Santos [14]. This multi-valuedness yields the desired white noise for the two-point functions (21) below, which would not be the case if the chemical potentials were single-valued.

A particular feature related to our choice above is that flipping signs of both $\epsilon$ and $\bar{\epsilon}$ can be absorbed in $\gamma_n$, which is not possible if only one of the signs is changed. We expect this behaviour to reflect in the disorder-averaged quantities we obtain in section 2.2 and later on.

An important characterisation of disorder in any system is given by the Harris criterion [17]. The criterion uses the mass dimension of the source $[\epsilon]$ to quantify whether a source of disorder is relevant ($[\epsilon] > 0$), marginal ($[\epsilon] = 0$) or irrelevant ($[\epsilon] < 0$). For a holographic scenario, the criterion can be obtained by considering the source term in the action of the dual field theory and using that the action is dimensionless in natural units. In our case, we consider the coupling term between the disordered source $\mu$ and its conserved current $\mathcal{L}$,

$$\int \mathrm{d}t \mathrm{d}\phi \, \mu \mathcal{L}. \tag{23}$$

If we were to consider the BTZ black hole instead of pure Poincaré patch, linear combinations of the zero modes of the currents $\mathcal{L}$ and $\bar{\mathcal{L}}$ would yield the black hole mass and angular momentum. Therefore we have $[\mathcal{L}] = 1$. In the above coupling term, this cancels the inverse mass dimension of $\mathrm{d}t$. Since the angular coordinate $\phi$ is dimensionless, we find $[\mu] = 0$ and correspondingly $[\epsilon] = 0$. This is consistent with the two-point function

$$\langle \mu(\phi)\mu(\theta) \rangle = 1 + \frac{\epsilon^2}{2}\delta(\phi - \theta). \tag{24}$$

Since $[\epsilon] = 0$, the disorder is marginal. To be more precise, we will find that the disorder we implement is marginally relevant, since after disordering and averaging the metric, there appear regions in the IR $z \to \infty$ where our perturbative treatment breaks down. This will be indicated e.g. by curvature singularities.

Having discussed the setup and its characteristics, we are now ready to analyse how the disorder affects the Poincaré patch geometry (15).

## 2.2 Averaged geometry

Using the setup described above, we proceed to analyse the disorder-averaged Poincaré patch. First, we compute the disorder-averaged metric as follows. We insert the chemical potentials (16) and (17) into the gauge field components (5), (6), (7) and (8), with $\mathcal{L}$ and $\bar{\mathcal{L}}$ given by (12) and (13), respectively, and setting $F = 0 = G$. The metric components then follow from (14). Expanding to second order and averaging[3] using the above mean values (22), we find

$$
\begin{aligned}
\langle g_{tt} \rangle &= \left(-1 + \frac{\epsilon\bar{\epsilon}}{2}\right)\frac{1}{z^2} + \frac{\epsilon\bar{\epsilon}}{12}, \\
\langle g_{t\phi} \rangle &= \frac{\bar{\epsilon}^2 - \epsilon^2}{24}, \\
\langle g_{\phi\phi} \rangle &= \frac{1}{z^2} + \frac{\epsilon^2 + \bar{\epsilon}^2}{24} - \frac{\epsilon\bar{\epsilon}}{40}z^2, \\
\langle g_{zt} \rangle &= 0 = \langle g_{z\phi} \rangle, \quad \text{and} \quad \langle g_{zz} \rangle = \frac{1}{z^2}.
\end{aligned}
\tag{25}
$$

As expected, to zeroth order in the disorder strengths this is the usual Poincaré patch metric. As mentioned above, the metric components are invariant under $(\epsilon, \bar{\epsilon}) \to -(\epsilon, \bar{\epsilon})$, but not under $(\epsilon, \bar{\epsilon}) \to (\epsilon, -\bar{\epsilon})$. The averaged geometry is in Fefferman–Graham form $\langle g_{zt} \rangle = \langle g_{z\phi} \rangle = 0$. For convenience, in the following calculations, we rescale $t$ such that the leading order in $\langle g_{tt} \rangle$ is equal to $-\frac{1}{z^2}$. Since we always work perturbatively in $\epsilon$ and $\bar{\epsilon}$, this does not change $\langle g_{t\phi} \rangle$.

---

[3]For completeness, the unaveraged components can be found in appendix A.

Regarding the curvature properties of the averaged metric (25), we first note that in the asymptotic region $z \to 0$, the metric describes a locally AdS spacetime with Ricci scalar $R = -6$, Ricci tensor squared $R_{\mu\nu}R^{\mu\nu} = 12$ and vanishing Cotton tensor. This is consistent with our choice of asymptotic boundary conditions. The above-mentioned rescaling of $t$ does not interfere with this observation.

Now we analyse the geometry described by the averaged metric for general $z$. Its Ricci scalar is given by

$$R = -6 + \frac{(\epsilon - \bar{\epsilon})^2}{12} z^2 + \frac{\epsilon\bar{\epsilon}}{10} z^4 . \tag{26}$$

In the bulk, a curvature singularity is present in the IR where $z \to \infty$. Also, the Cotton tensor has a non-vanishing component

$$C_{t\phi} = -\frac{3}{20} \epsilon\bar{\epsilon} z^2 , \tag{27}$$

with all other components equal to zero, so (25) in general differs from an AdS spacetime. Moreover, there is a singularity of the causal structure appearing at $g_{\phi\phi}(z_0) = 0$, with

$$z_0 = \left(\frac{40}{\epsilon\bar{\epsilon}}\right)^{\frac{1}{4}} + \mathcal{O}(\sqrt{\epsilon}) . \tag{28}$$

In the region $z > z_0$, as the $\phi$ direction is compact, the spacetime has closed timelike curves. This shows that at $z_0$ the spacetime ends, with finite Ricci curvature.

To further characterise the singularity, we analyse whether it yields a well-defined semiclassical approximation for a string [41]. If this is not the case the supergravity approximation that we employ is not valid anymore. To test this, we consider the string action for a small perturbation of a static configuration $z(\phi) = z_s(\phi) + \delta z(\phi)$,

$$\begin{aligned}
S &= \frac{1}{2\pi\alpha'} \int \mathrm{d}\phi \sqrt{(z')^2 + g_{\phi\phi}(z)} \\
&\approx \frac{1}{2\pi\alpha'} \int \mathrm{d}\phi \left( \sqrt{(z_s')^2 + g_{\phi\phi}(z_s)} + \frac{1}{2} \frac{(\delta z')^2}{\sqrt{(z_s')^2 + g_{\phi\phi}(z_s)}^3} \right) .
\end{aligned} \tag{29}$$

If the square root vanishes, the semiclassical approximation for the Nambu–Goto action (29) breaks down. For our averaged metric we find the real root $z_s = z_0$ and thus the supergravity approximation breaks down in the vicinity of $z_0$.

The appearance of singularities is not unexpected. They also appeared in prior work on disorder [14] and [22]. As in these cases, the disorder we use is a marginally relevant operator in the sense of the Harris criterion, therefore drastic changes in the interior of the bulk, i.e. the IR region, can arise. Since before averaging, the solutions for the charges in (12) and (13) correspond to a locally AdS$_3$ spacetime for any choice of $\mu(\phi), \bar{\mu}(\phi)$, the divergent behaviour must result from using the disorder-averaged metric. As briefly discussed in the introduction, the nature of the singularities found in [14, 22] was recently refined by a non-perturbative analysis in [18,19]. However, the setup of the latter two papers uses matter fields to source the disorder, which is conceptually different from our approach. Moreover, while they encounter Lifshitz scaling geometries, we do not find such a behaviour in our averaged metric (25). We therefore do not expect the methods developed in [18, 19] to have a direct relation to what we find in our approach. It will be interesting to study whether their approach can be adapted to our setup, which however we leave for future work.

Having discussed the properties of the averaged metric from the bulk perspective, in the next section we analyse the dual holographic state utilizing holographic renormalisation [44].

# 3 Properties of the holographic averaged state

In the previous section, we showed how the averaged metric is obtained, followed by a discussion of its curvature properties. Here, we study the effects of the disorder on the boundary field theory using the holographic dictionary. In particular, we calculate the boundary energy-momentum tensor and the QNEC.

## 3.1 Energy-momentum tensor and trace anomaly

Here, we compute the energy-momentum tensor of the dual theory on the boundary. Since the disorder-averaged metric (25) is in Fefferman–Graham gauge, we can do so by use of the standard result [44]

$$\langle\langle T_{ij}^{\text{ren}}\rangle\rangle = \frac{c}{12\pi}\left(\gamma_{ij}^{(2)} - \text{tr}\left(\gamma^{(2)}\right)\gamma_{ij}^{(0)}\right),\tag{30}$$

where $\gamma_{ij}$ is the metric induced on the boundary, expanded as

$$\gamma_{ij}(z) = \gamma_{ij}^{(0)}\frac{1}{z^2} + \gamma_{ij}^{(2)} + \dots\tag{31}$$

The trace is understood in terms of $\gamma_{ij}^{(0)}$ as $\text{tr}\left(\gamma^{(2)}\right) = \gamma^{(0)ij}\gamma_{ji}^{(2)}$. We denote the expectation value as $\langle\langle\cdot\rangle\rangle$ to distinguish it from disorder averages.

Applying (30) to the averaged metric in (25), we find the energy-momentum tensor

$$\langle\langle T_{ij}^{\text{ren}}[\langle\gamma\rangle]\rangle\rangle = \frac{c}{288\pi}\begin{pmatrix}\epsilon^2 + \bar\epsilon^2 & \bar\epsilon^2 - \epsilon^2 \\ \bar\epsilon^2 - \epsilon^2 & 2\epsilon\bar\epsilon\end{pmatrix}_{ij}.\tag{32}$$

It is clear from this result that by including disorder, we introduce a small amount of energy into the system. Provided that the disorder strengths $\epsilon$ and $\bar\epsilon$ are different, we also introduce angular momentum.

The trace of the energy-momentum tensor does not vanish in general,

$$\text{tr}\left(\langle\langle T_{ij}^{\text{ren}}[\langle\gamma\rangle]\rangle\rangle\right) = -\frac{c(\epsilon - \bar\epsilon)^2}{288\pi}.\tag{33}$$

However, since the disorder-averaged metric $\langle\gamma_{ij}^{(0)}\rangle = \text{diag}(-1, 1)$ induced on the boundary is Ricci flat, the general result for the trace anomaly in curved backgrounds

$$\text{tr}\left(\langle\langle T_{ij}^{\text{ren}}[\langle\gamma\rangle]\rangle\rangle\right) = \frac{c}{24\pi}R[\langle\gamma^{(0)}\rangle],\tag{34}$$

cannot be fulfilled for general disorder strengths $\epsilon$, $\bar\epsilon$. For equal strengths, both sides vanish. Even though $\text{tr}\left(\langle\langle T_{ij}^{\text{ren}}\rangle\rangle\right) \propto R[\langle\gamma^{(0)}\rangle]$ is fulfilled for $\epsilon = \bar\epsilon$, the bulk curvature (26) is still divergent in the IR, so these two features are not in one-to-one correspondence. Nevertheless, as we will show in section 4, there is a procedure to restore finite bulk curvature and absence of the trace anomaly for general $\epsilon$ and $\bar\epsilon$. Before doing so, we consider one additional holographic probe of our geometry, namely the QNEC.

## 3.2 Quantum null energy condition

QNEC locally constrains the null projections of the expectation value of the energy-momentum tensor [45–47]. For QFT$_2$ with a CFT$_2$ UV-fixed point it reads [45] (see also [48])

$$2\pi\langle\langle T_{ij}\rangle\rangle k^i k^j \geq S'' + \frac{6}{c}\left(S'\right)^2.\tag{35}$$

The right-hand side contains first and second variations of entanglement entropy (EE) explained below as well as the UV central charge $c$.

The variations of EE work as follows. One of the endpoints of the entangling interval is chosen arbitrarily, while the other must coincide with the spacetime point where the left-hand side of the QNEC inequality (35) is evaluated. Then the entangling region is deformed into the null direction $k^i$, with a deformation parameter $\lambda$. As a consequence of this lightlike deformation, EE depends on this parameter. Primes denote derivatives with respect to $\lambda$, and after taking these derivatives $\lambda$ is set to zero. For more details on QNEC in two dimensions and explicit examples see [49].

We investigate now QNEC for our averaged field theory, to verify whether or not QNEC holds/saturates/is violated. If the latter happened we had a nogo result showing that the averaged field theory cannot be a unitary relativistic QFT.

For states dual to three-dimensional vacuum Einstein solution QNEC saturates [50, 51], which in our context implies that QNEC saturates for each of our realisations. Therefore, also the averaged QNEC inequality trivially saturates,

$$\left\langle 2\pi \left\langle\langle T_{ij}\rangle\right\rangle k^i k^j \right\rangle = \left\langle S'' + \frac{6}{c}\left(S'\right)^2 \right\rangle. \tag{36}$$

The Cauchy–Schwarz inequality then implies a QNEC inequality for the averaged EE

$$\left\langle 2\pi \left\langle\langle T_{ij}\rangle\right\rangle k^i k^j \right\rangle \geq \langle S\rangle'' + \frac{6}{c}\left(\langle S\rangle'\right)^2. \tag{37}$$

However, in our work, we have to differentiate between averaged quantities and quantities computed from the averaged metric. Therefore, in the following, we calculate both sides of (35) explicitly for the averaged geometry (25).

We start our analysis by calculating the right-hand side of (35) for the averaged metric (25). The calculation follows the above description of introducing a lightlike deformation $k^i$ parametrised by a small $\lambda$. In the following, we choose the lightlike direction $k^i = (1, 1)$. The details of the calculation are included in appendix E, following the method described in [49]. From the computation using (25), we find

$$S''\big|_{\lambda=0} + \frac{6}{c}(S')^2\big|_{\lambda=0} = \frac{c}{3}\left[-\epsilon^2 + \left(2 - \frac{39}{25}l^2\right)\epsilon\bar{\epsilon} + 11\bar{\epsilon}^2\right], \tag{38}$$

where $l$ is the size of the undeformed entangling interval. The fact that this length appears proves that QNEC is not saturated, since $l$ cannot appear in the left-hand side of (35). Computing this explicitly from (32), we find

$$2\pi\langle\langle T_{ij}^{\mathrm{ren}}[\langle\gamma\rangle]\rangle\rangle k^i k^j = c\left(-\epsilon^2 + 2\epsilon\bar{\epsilon} + 3\bar{\epsilon}^2\right). \tag{39}$$

Comparing with (38), we find that whether QNEC holds or not depends on the length $l$ and the ratio of the disorder strengths $a = \frac{\bar{\epsilon}}{\epsilon}$. QNEC is satisfied within a symmetric region around $a = 1$ for sufficiently large $l$. When the disorder strengths are equal, QNEC always holds and even saturates for $l = 0$. Considering also the corresponding NEC condition, given by (39) via $\langle\langle T_{ij}^{\mathrm{ren}}\rangle\rangle k^i k^j \geq 0$, we find that there are regions in parameter space where NEC is violated while QNEC holds, as well as the contrary. We summarise our results in figure 1.

We point out that for the averaged geometry in (25), the EE shows unphysical behaviour in that it becomes negative for sufficiently large $l$; the explicit result follows from (E.12) in appendix E for $\lambda = 0$,

$$S = \frac{c}{3}\ln\frac{l}{z_{\mathrm{cut}}} + \frac{c}{864}\left[(\epsilon^2 + \bar{\epsilon}^2)l^2 - \frac{3\epsilon\bar{\epsilon}l^4}{25}\right]. \tag{40}$$

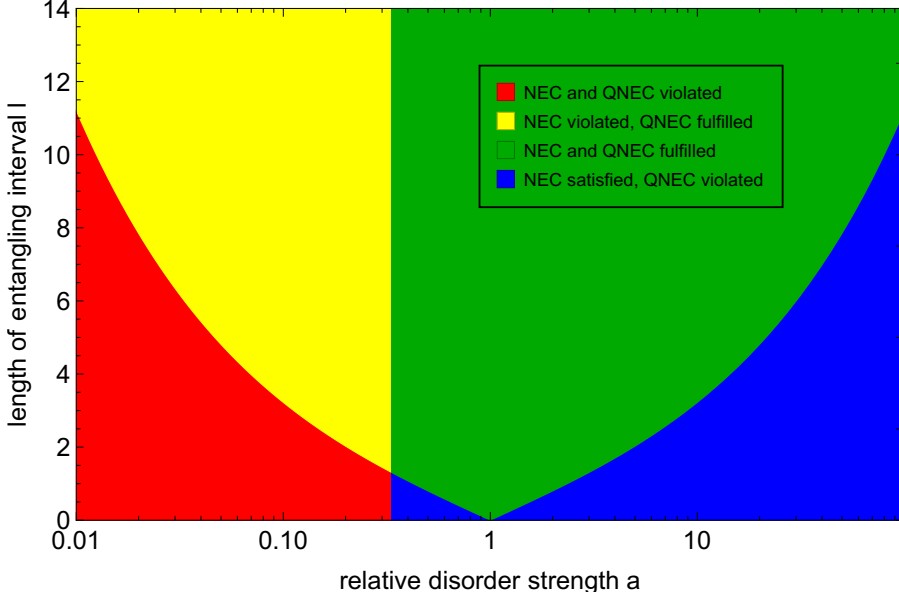

Figure 1: Results of the QNEC/NEC analysis for (25). All combinations of either QNEC or NEC being satisfied/violated exist for the averaged metric in (25).

The possibility for negative $S$ is related to the singular behaviour of (25) in the IR region. In particular, the turning point $z_s$ of the Ryu–Takayanagi geodesic comes close to the causal singularity $z_0$ obtained in (28), where our approximation breaks down. This feature of negative EE again shows that (25) includes unphysical behaviour, resulting from the disorder averaging. Motivated by the unphysical features associated with the averaged metric and its dual holographic quantities, in the next section we discuss a resummation method inspired by the Poincaré–Lindstedt resummation performed in earlier works on disorder [14, 22].

## 4 Poincaré–Lindstedt inspired resummation

As discussed in the previous section, the averaged metric has several unphysical properties. In this section, we resolve these deficiencies by a resummation method that we explain below.

In earlier works on disorder in holography such as [14, 22], it was observed that the averaged metric components contain secular terms, leading to Lifshitz scaling IR fixed points. To regulate the averaged solutions, the so-called Poincaré–Lindstedt resummation technique was employed. Inspired by this resummation, in the following we discuss an approach to modify the averaged Poincaré patch metric (25).

The idea for the resummation in our case is to change the averaged metric such that the curvature divergence is absent. To do so, we make an ansatz for the resummed metric, satisfying the following conditions:

- Since the divergence of the curvature is only in $z$, also the modifying functions should only depend on $z$.

- The divergent terms in (26) can be traced back to $\langle g_{tt} \rangle$ and $\langle g_{\phi\phi} \rangle$. Therefore only those components are modified.

- In the asymptotic region, $z \to 0$, the averaged metric is well behaved. Therefore, we demand the modifying functions to approach one for $z \to 0$.

While we implement these functions on the level of the averaged metric, we now argue that the resummation commutes with the averaging procedure: We define the resummation, independently of the level on that it is imposed, to only depend on the radial coordinate $z$, as this is the coordinate direction that shows the IR divergent behaviour. Furthermore, we do not want to alter the structure of the disorder in the field theory directions. If we now were to include the resummation functions before averaging, the functions could be pulled through the integrals involved in the averaging procedure (18). Therefore, we could also include the functions before the average without changing the result at the level of the averaged metric. For appropriately fixed resummation constants, the change of ansatz that this resummation amounts to will cure the issues in the averaged metric derived in sections 2.2 and 3.1, in particular the curvature singularities in the averaged metric. We stress again, that the change of ansatz done after averaging could as well have been implemented before. We implement the change of ansatz after averaging purely for calculational convenience. We elaborate on the implementation of this ansatz in the Chern–Simons formulation in section 6.

The above conditions lead us to the modified ansatz for the averaged metric

$$ds^2 = \frac{\langle g_{tt} \rangle}{\alpha(z)} dt^2 + \frac{dz^2}{z^2} + 2\langle g_{t\phi} \rangle dt d\phi + \frac{\langle g_{\phi\phi} \rangle}{\beta(z)} d\phi^2 \,, \tag{41}$$

where $\alpha$ and $\beta$ are functions of $z$ and $\epsilon, \bar{\epsilon}$. To satisfy the above conditions, we make a general ansatz

$$\alpha(z) = 1 + \sum_{n=1}^{\infty} a_n(\epsilon, \bar{\epsilon}) z^n \,, \qquad \beta(z) = 1 + \sum_{n=1}^{\infty} b_n(\epsilon, \bar{\epsilon}) z^n \,, \tag{42}$$

where the coefficients $a_n$, $b_n$ can be expanded[4] in $\epsilon$ and $\bar{\epsilon}$,

$$a_n(\epsilon, \bar{\epsilon}) = a_{1n}\epsilon^2 + a_{2n}\epsilon\bar{\epsilon} + a_{3n}\bar{\epsilon}^2 \,, \qquad b_n(\epsilon, \bar{\epsilon}) = b_{1n}\epsilon^2 + b_{2n}\epsilon\bar{\epsilon} + b_{3n}\bar{\epsilon}^2 \,. \tag{43}$$

In the following, the undetermined coefficients $a_{ij}$ and $b_{ij}$ will be fixed by recalculating the Ricci scalar for (41) to $\mathcal{O}(\epsilon^2)$ and demand the diverging terms to vanish. When expanding (41) to second order, the terms of $\mathcal{O}(z^n)$ in $\alpha(z)$ and $\beta(z)$, i.e., the coefficients $a_n(\epsilon, \bar{\epsilon})$ and $b_n(\epsilon, \bar{\epsilon})$, contribute at $\mathcal{O}(z^{n-2})$ in the metric. Therefore, we expect that only $a_2$, $b_2$, $a_4$ and $b_4$ need to be non-zero.

Computing the Ricci scalar for (41) to $\mathcal{O}(\epsilon^2)$ and demanding the result to be non-singular yields the following conditions:

$$
\begin{aligned}
b_{14} = b_{34} = a_{14} = a_{34} = 0 \,, & \qquad a_{24} + b_{24} = -\frac{1}{40} \,, \\
a_{12} + b_{12} = \frac{1}{24} = a_{32} + b_{32} \,, & \qquad a_{22} + b_{22} = -\frac{1}{12} \,.
\end{aligned}
\tag{44}
$$

As expected, all other coefficients have to vanish. Using the above relations to express $a_{ij}$ in terms of $b_{ij}$, the resummed metric (41) expanded to $\mathcal{O}(\epsilon^2)$ is given by

$$
\begin{aligned}
ds^2 = & \left[ -\frac{1}{z^2} + \left(\frac{1}{24} - b_{12}\right)\epsilon^2 - b_{22}\epsilon\bar{\epsilon} + \left(\frac{1}{24} - b_{32}\right)\bar{\epsilon}^2 - \left(\frac{1}{40} + b_{24}\right)z^2\epsilon\bar{\epsilon} \right] dt^2 \\
& + \left[ \frac{1}{z^2} + \left(\frac{1}{24} - b_{12}\right)\epsilon^2 - b_{22}\epsilon\bar{\epsilon} + \left(\frac{1}{24} - b_{32}\right)\bar{\epsilon}^2 - \left(\frac{1}{40} + b_{24}\right)z^2\epsilon\bar{\epsilon} \right] d\phi^2 \\
& + \frac{dz^2}{z^2} + 2\left(\frac{\bar{\epsilon}^2 - \epsilon^2}{24}\right) dt d\phi \,.
\end{aligned}
\tag{45}
$$

---

[4]This expansion may be generalised to higher order than quadratic straightforwardly.

By construction, the leading order terms in $z$ do not change. Hence, the averaged boundary metric is still given by two-dimensional Minkowski space.

The remaining undetermined parameters will be fixed in the following, but before we summarise what is achieved already by this resummation. To lowest order in $\epsilon$ the scalar curvature is now finite by construction, $R = -6$ for the entire range of $z$. Calculating the Ricci tensor for (45), we find that its trace-free part vanishes only provided that we set $b_{24} = -\frac{1}{40}$. This condition can also be found from related calculations: the three-dimensional Einstein equations with negative cosmological constant are satisfied only for this value of $b_{24}$. In addition, the Cotton tensor for the resummed metric does not vanish in general,

$$
C_{\mu\nu} = -(1 + 40 b_{24})\epsilon\bar{\epsilon}z^2 \begin{pmatrix} \frac{9}{80} & 0 & \frac{3}{20} \\ 0 & 0 & 0 \\ \frac{3}{20} & 0 & \frac{3}{16} \end{pmatrix}_{\mu\nu} ,
\tag{46}
$$

but only when fixing $b_{24} = -\frac{1}{40}$. That is, removing the $\mathcal{O}(z^2)$ terms in (45) ensures that (45) is locally an AdS spacetime everywhere.[5]

While the above features concern the bulk, the resummation also affects the dual holographic state. Since the terms of $\mathcal{O}(z^0)$ in $\langle g_{tt} \rangle$ and $\langle g_{\phi\phi} \rangle$ change, the boundary energy-momentum tensor is now given by

$$
\begin{aligned}
\langle\langle T_{tt}^{\text{ren}} \rangle\rangle &= \frac{c}{288\pi} \left[ (1 - 24 b_{12})\epsilon^2 - 24 b_{22}\epsilon\bar{\epsilon} + (1 - 24 b_{32})\bar{\epsilon}^2 \right] , \\
\langle\langle T_{t\phi}^{\text{ren}} \rangle\rangle &= \frac{c(\bar{\epsilon}^2 - \epsilon^2)}{288\pi} , \\
\langle\langle T_{\phi\phi}^{\text{ren}} \rangle\rangle &= \langle\langle T_{tt}^{\text{ren}} \rangle\rangle .
\end{aligned}
\tag{47}
$$

A particularly interesting feature is that the trace vanishes since the diagonal terms in (47) are equal. Therefore, by the resummation method, we obtain a metric with a dual energy-momentum tensor that satisfies the usual trace anomaly condition (34).

As apparent from the result above, the energy-momentum tensor now depends on the three remaining coefficients, $b_{12}$, $b_{22}$, and $b_{32}$. They enter in the diagonal terms, which can be thought of as a small mass induced by the disorder. Correspondingly, the off-diagonal terms are understood as small angular momentum.

In the following, we argue how to fix the open coefficients such that the least additional energy is injected into the system. In particular, we demand that the mass and angular momentum in the energy-momentum tensor are consistent with the BPS bound

$$
M^2 - J^2 \geq 0.
\tag{48}
$$

We start our argumentation by considering the limits where either $\epsilon = 0$ or $\bar{\epsilon} = 0$. Then, from (47), up to common constant coefficients, we find

$$
\text{for} \quad \bar{\epsilon} = 0: \quad M \propto (1 - 24 b_{12})\epsilon^2, \quad J \propto -\epsilon^2,
\tag{49}
$$

$$
\text{and for} \quad \epsilon = 0: \quad M \propto (1 - 24 b_{32})\bar{\epsilon}^2, \quad J \propto \bar{\epsilon}^2.
\tag{50}
$$

Inserting these values into (48), we obtain conditions on either $b_{12}$ or $b_{32}$,

$$
(1 - 24 b_{12})^2 - 1 \geq 0 \quad \rightarrow \quad \begin{cases} b_{12} \geq \frac{1}{12}, \\ b_{12} \leq 0, \end{cases}
\tag{51}
$$

$$
\text{and} \quad (1 - 24 b_{32})^2 - 1 \geq 0 \quad \rightarrow \quad \begin{cases} b_{32} \geq \frac{1}{12}, \\ b_{32} \leq 0. \end{cases}
\tag{52}
$$

---

[5]We find the same result if we had inserted the coefficients $b_{ij}$ in terms of $a_{ij}$. In this case, the terms $\propto z^2$ in the metric are proportional to an overall factor of $a_{24}$, which has to be set to zero to satisfy Einstein's equations. This is consistent with the constraint $a_{24} + b_{24} = -\frac{1}{40}$ and $b_{24} = -\frac{1}{40}$.

The upper conditions on $b_{12}$ and $b_{32}$ yield negative contributions to the mass. Therefore, we exclude them by further requiring $M \geq 0$. All values $b_{12} < 0$ and $b_{32} < 0$ yield a positive contribution to the mass. We now determine the resummation parameters such that the resummation procedure does not introduce any additional mass into the background spacetime, besides the mass introduced by the disorder. This amounts to setting $b_{12} = b_{32} = 0$. The expression for $M$ simplifies to

$$M \propto \epsilon^2 - 24\epsilon\bar{\epsilon}b_{22} + \bar{\epsilon}^2. \tag{53}$$

For this mass, we analyse again the BPS bound for general $\epsilon$ and $\bar{\epsilon}$. We find that (48) is satisfied when

$$a \leq 0 \wedge b_{22} \geq 0, \quad \text{or} \quad a \geq 0 \wedge b_{22} \leq 0, \tag{54}$$

where $a = \frac{\bar{\epsilon}}{\epsilon}$ is the ratio between the disorder parameters. The only choice for $b_{22}$ such that (48) holds for a generic ratio $a$ is $b_{22} = 0$. Therefore, with the parameters inserted in (47), we obtain

$$M \propto 1 + a^2, \quad \text{and} \quad J \propto -1 + a^2, \tag{55}$$

such that

$$\langle\langle T_{ij}^{\text{ren}}\rangle\rangle = \frac{c\epsilon^2}{288\pi} \begin{pmatrix} 1+a^2 & -1+a^2 \\ -1+a^2 & 1+a^2 \end{pmatrix}. \tag{56}$$

A particularly nice observation related to this choice of resummation parameters is that $\langle g_{\phi\phi}\rangle$ cannot have any real roots and does not change sign since it consists of a sum of manifestly positive terms. Therefore, also the closed timelike curves are removed by the resummation. Moreover, also the semiclassical approximation is well-defined for a string (29) on the resummed metric.

There also exists a different approach than the above procedure to fix the resummation coefficients $a_{ij}$ and $b_{ij}$, which remarkably leads to the same result: Inserting the resummation ansatz (41) into the Einstein equations yields differential equations on the functions $\alpha(z)$ and $\beta(z)$. Expanding these functions as $\alpha(z) = 1 + \epsilon^2\alpha_2(z) + \mathcal{O}(\epsilon^4)$, $\beta(z) = 1 + \epsilon^2\beta_2(z) + \mathcal{O}(\epsilon^4)$ simplifies the differential equations to

$$0 = \beta_2''(z) - \frac{\beta_2'(z)}{z} + \frac{az^2}{5}, \tag{57}$$

$$0 = \alpha_2'(z) + \beta_2'(z) - \frac{z}{12}(1 - 2a + a^2) + \frac{az^3}{10}, \tag{58}$$

$$0 = \alpha_2''(z) - \frac{\alpha_2'(z)}{z}, \tag{59}$$

at quadratic order in $\epsilon$, where $'$ denotes a derivative with respect to $z$. The solutions to the differential equations for $\alpha_2(z)$ and $\beta_2(z)$ fix the metric to read

$$ds^2 = \left[-\frac{1}{z^2} + \frac{\epsilon^2 + \bar{\epsilon}^2}{24} - \frac{c_1}{2}\epsilon^2\right]dt^2 + \frac{dz^2}{z^2}$$
$$+ \left[\frac{1}{z^2} + \frac{\epsilon^2 + \bar{\epsilon}^2}{24} - \frac{c_1}{2}\epsilon^2\right]d\phi^2 + 2\left(\frac{\bar{\epsilon}^2 - \epsilon^2}{24}\right)dt d\phi, \tag{60}$$

where $c_1$ is an integration constant that will be fixed shortly by using the BPS bound.[6] By this procedure, we directly arrive at the resummed metric with fixed coefficients $b_{i2} = 0$, which

---

[6]Note that by definition, $c_1$ is independent of $\epsilon$. It does, however, depend on $a$, such that the expression in (60) is symmetric in $\epsilon \leftrightarrow \bar{\epsilon}$. To be more specific, the expression $c_1 = \xi_1 + a\xi_2 + a^2\xi_3$ has to satisfy $\xi_1 = \xi_3$, for $\epsilon^2 c_1$ to be symmetric under $\epsilon \leftrightarrow \bar{\epsilon}$.

coincides with the above result of the BPS bound analysis, up to the constant term $-\frac{c_1}{2}$. Again invoking the BPS bound, now for (60), we find that $c_1 \geq 0$, so we can consistently set it to zero, ending up with the same metric as by the previous analysis.

Although these are promising results, we point out that the above description has certain differences from the method used in [14]. The most pertinent is that the Poincaré–Lindstedt method was employed in [14] *before* averaging while we choose to change the effective metric *after* averaging. Furthermore, while we encounter IR divergences in curvature invariants, in their case only the metric components are IR-divergent while curvature remains finite.

Having discussed some basic geometric quantities as well as boundary properties of the resummed metric, in the remainder of this section, we analyse the effect of the resummation on QNEC.

Since the resummation leads to an averaged geometry that is a solution to the AdS vacuum Einstein equations, we also expect a change in the result for QNEC. Doing the calculation for (45), we find that the right-hand side of (35) is given by

$$
S''\big|_{\lambda=0} + \frac{6}{c}\left(S'\right)^2\big|_{\lambda=0} = \frac{c}{3}\left[-b_{12}\epsilon^2 - b_{22}\epsilon\bar{\epsilon} + \left(\frac{1}{12} - b_{32}\right)\bar{\epsilon}^2\right].
\tag{61}
$$

The energy-momentum tensor after resummation is given in (47). Projecting onto the lightlike direction $k^i(1,1)$ yields

$$
2\pi\langle\langle T_{ij}^{\mathrm{ren}}\rangle\rangle k^i k^j = \frac{c}{3}\left[-b_{12}\epsilon^2 - b_{22}\epsilon\bar{\epsilon} + \left(\frac{1}{12} - b_{32}\right)\bar{\epsilon}^2\right],
\tag{62}
$$

so (35) saturates independently of $a = \frac{\bar{\epsilon}}{\epsilon}$, $l$, or the remaining parameters of the resummation. This result is consistent with the fact that the resummed geometry is a solution to the vacuum Einstein equations and thereby a Bañados geometry, since for all Bañados geometries QNEC saturates [49].

In particular, this means that also the values $b_{12} = b_{22} = b_{32} = 0$ are allowed, for which we argued above. In this case, also NEC given by (62) via $\langle\langle T_{ij}^{\mathrm{ren}}\rangle\rangle k^i k^j \geq 0$ is satisfied.

As a final comment, after resummation, the EE is positive and receives only positive perturbative corrections. Explicitly, it is given by (E.12) for $\lambda = 0$ without the term $\propto \epsilon\bar{\epsilon}l^4$,

$$
S = \frac{c}{3}\ln\frac{l}{z_{\mathrm{cut}}} + \frac{c(\epsilon^2 + \bar{\epsilon}^2)l^2}{864}.
\tag{63}
$$

# 5 Comparing other approaches to averaging

In the previous section, we have shown in the metric formulation how the deficiencies of the averaged metric can be cured by a resummation procedure inspired by the Poincaré–Lindstedt method. These deficiencies, in particular the divergence of the Ricci scalar in the IR region of the bulk, are not only present perturbatively to order $\mathcal{O}(\epsilon^2)$, but appear also to fourth order and even non-perturbatively (see app. C and app. D, respectively). In the following, we compare the above resummation approach to two alternative approaches of studying the disordered setup. First, we average directly in the Chern–Simons formulation. Second, we discuss the differences of the above to averaging only after calculating the quantity of interest.

## 5.1 Averaged connection and resummation in Chern–Simons

The original sources of the disorder are the chemical potentials of the Chern–Simons formulation, (16) and (17). As we pointed out earlier in section 2, directly averaging the connections $A$ and $\bar{A}$ will, in general, not result in the averaged metric obtained in section 2.2 due to the

non-linear relation between metric and gauge field (14). To make a quantitative comparison, in the following, we directly average the connections. After stating the results of this average, we will also discuss how a resummation in the Chern–Simons formulation may be performed.

The components of the connections are given in (5), (6), (7) and (8). Inserting the chemical potentials (16) and (17), with $\mathcal{L}$ and $\bar{\mathcal{L}}$ given by (12) and (13) for $F = 0 = G$, the average is straightforward to perform using the mean values (22). Note that, except for $\mathcal{L}$ and $\mu\mathcal{L}$ as well as their barred analogues, all other averages are trivially performed. The resulting averaged connection is given by

$$a_t = T_+ + \frac{\epsilon^2}{24}T_-, \qquad a_\phi = T_+ - \frac{\epsilon^2}{24}T_-, \tag{64}$$

$$\bar{a}_t = -\frac{\bar{\epsilon}^2}{24}T_+ - T_-, \qquad \bar{a}_\phi = -\frac{\bar{\epsilon}^2}{24}T_+ + T_-. \tag{65}$$

To zeroth order in the disorder strength, this is simply the Chern–Simons version of the Poincaré patch. By the disorder, the respective lowest components ($T_-$ for $A$, $T_+$ for $\bar{A}$) of the connections receive corrections. In particular, these corrections are such that the gauge flatness conditions (9) are no longer satisfied,

$$dA + A \wedge A = -\frac{\epsilon^2}{6}T_0 dt \wedge d\phi, \quad \text{and} \quad d\bar{A} + \bar{A} \wedge \bar{A} = -\frac{\bar{\epsilon}^2}{6}T_0 dt \wedge d\phi. \tag{66}$$

Calculating the metric for the averaged connections, we find

$$g_{tt}\left(\langle A \rangle, \langle \bar{A} \rangle\right) = -\frac{1}{z^2} - \frac{\epsilon^2 + \bar{\epsilon}^2}{24}, \quad g_{t\phi}\left(\langle A \rangle, \langle \bar{A} \rangle\right) = 0, \tag{67}$$

$$g_{zz}\left(\langle A \rangle, \langle \bar{A} \rangle\right) = \frac{1}{z^2}, \quad \text{and} \quad g_{\phi\phi}\left(\langle A \rangle, \langle \bar{A} \rangle\right) = -g_{tt}\left(\langle A \rangle, \langle \bar{A} \rangle\right). \tag{68}$$

The Ricci scalar of this metric shows a divergence in the IR,

$$R = -6 + \frac{\epsilon^2 + \bar{\epsilon}^2}{6}z^2. \tag{69}$$

While the quantitative behaviour of the Ricci scalar is different compared to (26), the qualitative behaviour is similar, in the sense that an IR divergence is present.

In the dual description, the boundary metric is given by the Minkowski metric $\eta_{ij}$. The boundary energy-momentum tensor following from (68) is given by

$$\langle\langle T_{ij}^{\text{ren}} \rangle\rangle = \frac{c}{288\pi}\begin{pmatrix} \epsilon^2 + \bar{\epsilon}^2 & 0 \\ 0 & -\epsilon^2 - \bar{\epsilon}^2 \end{pmatrix}_{ij} = -\frac{c(\epsilon^2 + \bar{\epsilon}^2)}{288\pi}\eta_{ij}. \tag{70}$$

Since the boundary is flat, the relation for the trace anomaly in curved backgrounds (34) cannot be satisfied.

To cure these issues, we perform a resummation in the Chern–Simons formulation. We do so by modifying the lowest weight components of $a_t$ and $\bar{a}_t$. Explicitly, we add free constants $\tilde{\zeta} = \epsilon^2\zeta$ and $\tilde{\bar{\zeta}} = \bar{\epsilon}^2\bar{\zeta}$ which vanish when the disorder strengths are set to zero. The resummed connection components are then given by

$$a_t^{\text{res}} = T_+ + \left(\frac{1}{24} + \zeta\right)\epsilon^2 T_-, \tag{71}$$

$$\bar{a}_t^{\text{res}} = \left(-\frac{1}{24} + \bar{\zeta}\right)\bar{\epsilon}^2 T_+ - T_-. \tag{72}$$

For these connections, the gauge flatness conditions (9) evaluate to

$$\mathrm{d}A + A \wedge A = -\frac{\epsilon^2}{6}(1 + 12\zeta)T_0 \mathrm{d}t \wedge \mathrm{d}\phi \,, \tag{73}$$

$$\mathrm{d}\bar{A} + \bar{A} \wedge \bar{A} = -\frac{\bar{\epsilon}^2}{6}(1 - 12\bar{\zeta})T_0 \mathrm{d}t \wedge \mathrm{d}\phi \,. \tag{74}$$

Demanding that the resummed connections are gauge flat fixes the resummation constants to

$$\zeta = -\frac{1}{12} = -\bar{\zeta} \,. \tag{75}$$

An alternative way to fix these constants is to calculate the energy-momentum tensor dual to the metric following from the resummed connections with open $\zeta, \bar{\zeta}$. Demanding that this energy-momentum is traceless such that (34) is satisfied yields the same values for $\zeta, \bar{\zeta}$ as given in (75).

Inserting these values, the resummed connection components are given by

$$a_t^{\mathrm{res}} = T_+ - \frac{\epsilon^2}{24}T_- \,, \tag{76}$$

$$\bar{a}_t^{\mathrm{res}} = \frac{\bar{\epsilon}^2}{24}T_+ - T_- \,. \tag{77}$$

In general, calculating the metric from the gauge fields and averaging do not commute. Remarkably, the metric resulting from the resummed connections,

$$\mathrm{d}s^2 = \left[-\frac{1}{z^2} + \frac{\epsilon^2 + \bar{\epsilon}^2}{24}\right]\mathrm{d}t^2 + \frac{\mathrm{d}z^2}{z^2} + \left[\frac{1}{z^2} + \frac{\epsilon^2 + \bar{\epsilon}^2}{24}\right]\mathrm{d}\phi^2 + 2\left(\frac{\epsilon^2 - \bar{\epsilon}^2}{24}\right)\mathrm{d}t\mathrm{d}\phi \,, \tag{78}$$

is the same that we found in the previous section when discussing the resummation on the level of the averaged metric, up to a sign change in the $t\phi$ component. This component corresponds to the angular momentum. As discussed at the end of the introduction, we may remove this sign difference by $t \to -t$. The BPS condition analysed in the previous section is quadratic in the angular momentum, so it is still satisfied. Also, analogous to before, the metric (78) has no curvature singularities and is locally an AdS spacetime everywhere. The dual energy-momentum tensor is traceless and, therefore, satisfies (34).

## 5.2 Calculating before averaging

Alternatively to the approaches in the previous sections, in the following we compare the above results to first calculating observables for each realisation of the disorder and averaging only the final result. As we mentioned above, computing and averaging does not commute due to the non-linearity of both the average itself as well as the non-linear dependence of certain observables on the chemical potentials as sources of the disorder. We will make this explicit by computing the boundary energy-momentum tensor for each realisation and discussing its properties upon averaging the result.

Here we perform the same computations as in section 3.1 for the expanded, but not averaged, metric components displayed in (A.4). Adjusting (30) to the case where the metric is not in Fefferman–Graham gauge, we obtain the energy-momentum tensor in each disorder realisation. To properly compare it to our previous results, we average the expression to obtain

$$\langle\langle T_{ij}^{\mathrm{ren}}[\gamma]\rangle\rangle = -\frac{c}{288\pi}\begin{pmatrix} \frac{3}{2}(\epsilon - \bar{\epsilon})^2 & \bar{\epsilon}^2 - \epsilon^2 \\ \bar{\epsilon}^2 - \epsilon^2 & \frac{1}{2}(\epsilon - \bar{\epsilon})^2 \end{pmatrix}. \tag{79}$$

This result differs from both the energy-momentum tensor obtained directly from the averaged metric as well as the energy-momentum tensor after resummation, displayed in (32) and (56), respectively. In particular, for equal disorder $\epsilon = \bar{\epsilon}$ all components are vanishing. To test whether the above energy-momentum tensor satisfies the trace anomaly equation, we have to also compute the scalar curvature of the disorder boundary metric $\gamma_{ij}^{(0)}$. Since in this section, we average only after every other computational step, the boundary metric depends on the angular coordinate $\phi$ due to the disorder. Correspondingly, its Ricci scalar does not vanish,

$$\langle R[\gamma_{ij}^{(0)}]\rangle = -\frac{(\epsilon - \bar{\epsilon})^2}{12}\,.\tag{80}$$

To compute the trace of the energy-momentum tensor of the boundary theory, it is important to not compute the trace of (79) using the average of the boundary metric, but to compute the trace for each realisation and averaging afterwards. This illustrates the non-commutative behaviour of averaging and computing in

$$\mathrm{tr}\left(\langle\langle T_{ij}^{\mathrm{ren}}[\gamma]\rangle\rangle\right) = \langle\gamma^{ij}\rangle\langle\langle T_{ji}^{\mathrm{ren}}[\gamma]\rangle\rangle \neq \left\langle\mathrm{tr}\left(T_{ij}^{\mathrm{ren}}[\gamma]\right)\right\rangle.\tag{81}$$

Having this in mind, averaging the resulting expression for the trace of the energy-momentum tensor yields

$$\langle\mathrm{tr}\left(T_{ij}^{\mathrm{ren}}[\gamma]\right)\rangle = -\frac{c(\epsilon - \bar{\epsilon})^2}{288\pi}\,.\tag{82}$$

Combining with (80), this satisfies the trace anomaly equation (34).

The non-commuting behaviour (81) makes manifest a particular disadvantage of averaging only after computing the quantity of interest. That is, it is not straightforward to associate an effective metric to the energy-momentum tensor (79), in particular not by averaging the boundary metric. Rather, while the boundary metric $\gamma_{ij}^{(0)}$ is crucial for computing quantities such as traces or more generally contracting indices, it does not provide an effective description of the boundary spacetime corresponding to the averaged boundary energy-momentum tensor (79).

## 6 Conclusions and outlook

In this paper, we studied the effect of random disorder on the Poincaré patch solution of the three-dimensional Einstein equations for the AdS vacuum. Starting from the Chern–Simons formulation of AdS$_3$ gravity with Brown–Henneaux boundary conditions, we used the left- and right-moving chemical potentials $\mu$ and $\bar{\mu}$ to introduce disorder. Inspired by earlier work on disorder in holography, we chose a convenient ansatz for the disorder potential and worked perturbatively to second order in the disorder strengths $\epsilon$ and $\bar{\epsilon}$. This enabled us to perform an analytic average. Before discussing our findings, it should be noted that this choice of implementing the disorder is not unique. An alternative approach would be to directly disorder the boundary metric on which the dual 2D CFT lives. It is not obvious whether both approaches lead to the same results.

Having obtained the averaged metric, we first studied its bulk properties. As expected from our choice of boundary conditions, in the ultraviolet region $z \to 0$ we maintained asymptotic AdS behaviour, i.e., the Ricci scalar was constant and equal to $-6$. Furthermore, the trace-free Ricci tensor as well as the Cotton tensor vanished asymptotically. Deeper in the bulk, we found that this is no longer true. All the quantities mentioned before receive corrections quadratic in $\epsilon$ and $\bar{\epsilon}$. These corrections depended on the holographic coordinate $z$. This resulted in an unphysical curvature divergence and a corresponding naked singularity in the IR region

$z \to \infty$, implying a breakdown of our perturbative treatment. Such divergent behaviour was not unexpected since similar effects appeared also in earlier work on disorder [14, 22]. Moreover, by the Harris criterion, the disorder we used is marginally relevant. Therefore, large changes in the IR region are expected to happen. The marginally relevant nature of our disorder is ultimately responsible for the breakdown of perturbation theory. We further found that the averaged metric did not yield a well-defined semiclassical approximation for a string, which is related to the fact that the averaged metric component $\langle g_{\phi\phi} \rangle$ has real roots $z_0$. This also implied that the averaged metric had a singularity in the causal structure at $z_0$. Since the $\phi$-direction is compact, the region $z > z_0$ includes closed timelike curves, indicating that the spacetime ends at $z_0$ with finite Ricci curvature.

We also studied the holographic state dual to the averaged metric. Computing the energy-momentum tensor, we found a non-vanishing trace. Since the metric induced on the boundary is just two-dimensional Minkowski space, the relation between trace anomaly and boundary curvature was not fulfilled. We also analysed the entanglement properties of the averaged geometry. We found that the EE can become negative for a sufficiently large length of the entangling region $l$ in the boundary theory, related to the breakdown of perturbation theory in the bulk IR region. Again, this is due to the marginally relevant nature of the disorder. Furthermore, we computed QNEC for the averaged metric. It depended on the length $l$ and the ratio of the disorder strengths $a$ whether or not it was fulfilled. Moreover, we also analysed NEC and found that it can be violated as well, see Figure 1.

In earlier works on disorder, to regulate diverging behaviour, the so-called Poincaré–Lindstedt method was employed. Inspired by this, we use an analogous method to cure the above-mentioned deficiencies of the averaged geometry. Although we modify only the averaged geometry, the averaging procedure and our resummation method commute since the resummation functions do not depend on the random phases sourcing the disorder. We rescaled the averaged metric components by the resummation functions containing undetermined coefficients. By computing quantities such as the Ricci curvature and the trace-free Ricci tensor, we obtain constraints on these coefficients. By our method, we were able to restore finite curvature throughout the entire bulk. The resummed metric was a solution to the AdS$_3$ vacuum Einstein equations with vanishing Cotton tensor. Furthermore, the closed timelike curves were removed and the semiclassical approximation for a string could be defined. In the dual theory, the energy-momentum tensor on the boundary was traceless, such that the conformal anomaly equation was satisfied. The EE was manifestly positive and QNEC saturated. Three of the resummation coefficients entered the boundary energy-momentum tensor of the resummed metric. We fixed them to zero by requiring consistency with the BPS bound for generic values of $\epsilon$ and $\bar{\epsilon}$.

Apart from the metric formalism, we also studied the averaging procedure directly in the Chern–Simons formulation, i.e. averaging the connections. After averaging, the flatness conditions are no longer satisfied. The metric calculated from the averaged connections shows a divergence in the IR as well, although the behaviour of the divergence is different than in the metric approach. In the dual picture, the trace of the boundary energy-momentum tensor derived from the averaged connections does not vanish and in particular, does not satisfy the relation between trace anomaly and boundary curvature.

As in the metric approach, we performed a resummation to cure these issues. In particular, we modified the lowest weight components of $a_t$ and $\bar{a}_t$ by free parameters proportional to $\epsilon^2$ and $\bar{\epsilon}^2$, respectively. Demanding that the flatness conditions are satisfied fixes these constants uniquely. All the aforementioned issues are cured by this resummation. The curvature singularity in the IR is removed and the relation between trace anomaly and boundary curvature is satisfied. Interestingly, the metric obtained from the resummed connections is, up to an overall sign in the $t\phi$-component, the same as the one obtained by the resummation of

the averaged metric, although calculating the metric from the connections and averaging, in general, do not commute in the Chern–Simons formalism as well. Using a time reversal transformation $t \to -t$, this sign difference is removed. This is yet another instance of averaging and disordering not commuting, this time between the Chern–Simons and metric formalisms, which are equivalent to each other within each disorder realisation before averaging but are found to differ by a sign after averaging.

Furthermore, as discussed in section 5.2, also in the metric formalism, the non-commuting of disordering and averaging manifests itself in the calculation of observables: For example, if we first calculate the energy-momentum tensor expectation value in each realization and then average, it is not straightforward to associate to the resulting boundary energy-momentum tensor (79) an effective metric which would source this energy-momentum tensor. In particular, the averaged boundary metric does not source the energy-momentum tensor (79), which vanishes for example in the limit $\epsilon = \bar{\epsilon}$, while the averaged boundary metric read off from (25) does not vanish. This implies that the boundary metric $\gamma^{(0)}_{ij}$ does not provide an effective description of the boundary spacetime corresponding to the averaged boundary energy-momentum tensor (79).

Our analysis conducted in this work enables numerous future research projects. In the following, we give a list of questions and issues that we have not answered or studied in this work but which might be interesting to study in the future.

**Effective theory.** Before averaging the metric, the dynamics of the theory is determined by the Einstein–Hilbert action. After averaging, this is no longer true. The question arises if one can find an effective action where the averaged metric follows as a solution to the equations of motion. We made two such attempts by considering topologically massive gravity (TMG) and Einstein-dilaton gravity. For TMG, the Chern–Simons term yielding the non-vanishing graviton mass has a coupling constant $\mu$. Imposing the Hamiltonian constraint, Eq. (4) in [52], we found that the resulting $\mu$ does not have a constant asymptotic piece. Therefore, (25) cannot be a solution to TMG [52]. To check Einstein-dilaton gravity, we followed the steps discussed in [53]. To calculate the dilaton field from the metric, the averaged metric components need to satisfy certain convexity conditions. Unfortunately, the averaged metric does not satisfy these conditions. Finding a bulk effective theory would significantly improve the understanding of the disorder averaging in AdS/CFT and, in particular, allow us to study the dual field theory directly.

**Poincaré–Lindstedt inspired resummation.** Regarding the resummation method, an interesting question is tied to the effective theory. Above we discussed in great detail how the resummation regulates the averaged geometry. If we had an effective theory at hand, it would be interesting to study how it affects the resummation.

**Disordering different background geometries.** For simplicity, in the current work we restricted our focus on the Poincaré patch metric. On top of this, we introduced disorder. From the first-order formulation of AdS$_3$ gravity that we introduced in section 2, it is straightforward to extend to other cases. Simple examples are the BTZ black hole, global AdS$_3$, or conical defect geometries. These examples are obtained simply by fixing the functions $F$ and $G$ in (12) and (13) to be non-vanishing constants. It will be interesting to see if there are qualitatively different features after averaging the corresponding disordered metrics.

**Beyond perturbation theory.** Throughout the entire paper, we worked perturbatively in the disorder strengths $\epsilon$ and $\bar{\epsilon}$. While this enables us to perform the averages analytically, it begs the question of to what extent the perturbative treatment affects the result. One might

expect that in a non-perturbative treatment, some of the problematic features are absent. Since the non-perturbative calculation involves averages of inverse powers of $\mu$ and $\bar{\mu}$, an analytic answer to this question is hard to obtain. However, a numeric analysis, along the lines of the numerical non-perturbative evaluation of the Ricci scalar outlined in section D will be very useful to properly address these questions.

**Different boundary conditions.** In our procedure, we average over descendant geometries of Bañados type, which include a conical singularity in the IR. It is known that by relaxing the Brown–Henneaux boundary conditions to e.g. near horizon boundary conditions, this conical singular behaviour can be removed [54, 55]. It is reasonable to assume that different boundary conditions might lead to an averaged metric with regular curvature. To answer this question, but also to study disorder in $2D$ CFTs with extended symmetries [40], it is interesting to study disorder averages with different boundary conditions.

# Acknowledgements

We thank Souvik Banerjee, Johanna Erdmenger, Simeon Hellerman, Ioannis Papadimitriou and Shahin Sheikh-Jabbari for useful discussions.

M. D., R. M. and S. Z. are grateful for the kind hospitality during their stay at TU Wien where a part of this project was realised.

**Funding information** M. D., R. M. and S. Z. acknowledge support by the Deutsche Forschungsgemeinschaft (DFG, German Research Foundation) under Germany's Excellence Strategy through the Würzburg-Dresden Cluster of Excellence on Complexity and Topology in Quantum Matter - ct.qmat (EXC 2147, project-id 390858490). Their work is furthermore supported via project-id 258499086 - SFB 1170 'ToCoTronics'. D. G. was supported by the Austrian Science Fund (FWF), projects P 30822, P 32581 and P 33789. The final part of this research was conducted while DG was visiting the Okinawa Institute of Science and Technology (OIST) through the Theoretical Sciences Visiting Program (TSVP). S. Z. is financially supported by the China Scholarship Council.

# A Expanded metric components

In this appendix, we display the metric components resulting from the gauge fields expressed in terms of the charges $\mathcal{L}$ and $\bar{\mathcal{L}}$ as well as the chemical potentials $\mu, \bar{\mu}$. To calculate the metric components, we use the following basis for $\mathfrak{sl}(2, \mathbb{R})$,

$$T_+ = \begin{pmatrix} 0 & 0 \\ 1 & 0 \end{pmatrix}, \quad T_0 = \frac{1}{2}\begin{pmatrix} 1 & 0 \\ 0 & -1 \end{pmatrix}, \quad \text{and} \quad T_- = \begin{pmatrix} 0 & -1 \\ 0 & 0 \end{pmatrix}. \tag{A.1}$$

The trace of two generators is given by

$$\mathrm{tr}(T_a T_b) = \begin{pmatrix} 0 & 0 & -1 \\ 0 & \frac{1}{2} & 0 \\ -1 & 0 & 0 \end{pmatrix}, \tag{A.2}$$

where we ordered $(+, 0, -)$.

The metric components follow from inserting (5), (6), (7), (8) and the group element $b = \exp(\rho T_0)$ into (14). Expressed in the holographic coordinate $z = \exp(-\rho)$, we find

$$g_{tt} = \frac{\mu\bar{\mu}}{z^2} + \frac{2\pi}{k}\left(\mu^2\mathcal{L} - \bar{\mu}^2\bar{\mathcal{L}}\right) + \frac{1}{4}\left(\mu' + \bar{\mu}'\right)^2 - \frac{1}{2}\left(\mu\mu'' + \bar{\mu}\bar{\mu}''\right)$$
$$+ \left(\frac{\mu''\bar{\mu}''}{4} - \frac{\pi}{k}\left(\mathcal{L}\mu\bar{\mu}'' - \bar{\mathcal{L}}\bar{\mu}\mu''\right) - \frac{4\pi^2}{k^2}\mathcal{L}\mu\bar{\mathcal{L}}\bar{\mu}\right)z^2, \tag{A.3a}$$

$$g_{tz} = \frac{\mu' + \bar{\mu}'}{2z}, \tag{A.3b}$$

$$g_{t\phi} = \frac{\mu + \bar{\mu}}{2z^2} + \frac{2\pi}{k}\left(\mathcal{L}\mu - \bar{\mathcal{L}}\bar{\mu}\right) - \frac{\mu'' + \bar{\mu}''}{4} - \left(\frac{\pi}{2k}\left(\mathcal{L}\bar{\mu}'' - \bar{\mathcal{L}}\mu''\right) + \frac{2\pi^2}{k^2}\mathcal{L}\bar{\mathcal{L}}\left(\mu + \bar{\mu}\right)\right)z^2, \tag{A.3c}$$

$$g_{zz} = \frac{1}{z^2}, \tag{A.3d}$$

$$g_{z\phi} = 0, \tag{A.3e}$$

$$g_{\phi\phi} = \frac{1}{z^2} + \frac{2\pi}{k}\left(\mathcal{L} - \bar{\mathcal{L}}\right) - \frac{4\pi^2}{k^2}\mathcal{L}\bar{\mathcal{L}}z^2. \tag{A.3f}$$

Inserting (12), (13) and subsequently (16) and (17), we expand to second order in $\epsilon$ and $\bar{\epsilon}$. We find the following metric components:

$$g_{tt} = \left(-1 + (\bar{\epsilon} - \epsilon)f + \epsilon\bar{\epsilon}f^2\right)\frac{1}{z^2} + \frac{\epsilon\bar{\epsilon}}{2}f'^2, \tag{A.4a}$$

$$g_{tz} = \frac{(\bar{\epsilon} + \epsilon)f'}{2z}, \tag{A.4b}$$

$$g_{t\phi} = \frac{\epsilon + \bar{\epsilon}}{2}f\frac{1}{z^2} + \frac{\bar{\epsilon}^2 - \epsilon^2}{4}f'^2 + \frac{\epsilon + \bar{\epsilon}}{4}f'', \tag{A.4c}$$

$$g_{zz} = \frac{1}{z^2}, \tag{A.4d}$$

$$g_{z\phi} = 0, \tag{A.4e}$$

$$g_{\phi\phi} = \frac{1}{z^2} - \frac{\bar{\epsilon}^2 + \epsilon^2}{2}ff'' - \frac{\bar{\epsilon}^2 + \epsilon^2}{4}f'^2 + \frac{\epsilon - \bar{\epsilon}}{2}f'' - \frac{\epsilon\bar{\epsilon}}{4}f''^2z^2. \tag{A.4f}$$

Note that these components are expanded only in the disorder strengths $\epsilon, \bar{\epsilon}$; there is no expansion involved in $z$.

## B  Disorder with relative phase shift

In this appendix, we discuss the effect of a more general ansatz for the disorder functions. As mentioned in the main text, we could allow for random phases $\gamma_n$ and $\bar{\gamma}_n$ with a relative phase shift in the intervals that they are drawn from, $[0, 2\pi)$ and $[\alpha, 2\pi + \alpha)$, respectively. Compared to (16) and (17), this relative phase shift in the intervals results in a relative phase shift in the cosines,

$$f(\phi) = \frac{1}{\sqrt{N}}\sum_{n=1}^{N}\cos\left(\frac{n}{N}\phi + \gamma_n\right), \quad \text{and} \quad \bar{f}(\phi) = \frac{1}{\sqrt{N}}\sum_{n=1}^{N}\cos\left(\frac{n}{N}\phi + \gamma_n + \alpha\right). \tag{B.1}$$

Using this ansatz, we first observe that all of the one-point functions still vanish,

$$\langle f \rangle = \langle \bar{f} \rangle = 0. \tag{B.2}$$

Moreover, all disorder averages without mixing of $f$ and $\bar{f}$ are left invariant,

$$\langle f^2 \rangle = \langle \bar{f}^2 \rangle = \frac{1}{2}, \qquad \langle (f')^2 \rangle = \langle (\bar{f}')^2 \rangle = \frac{1}{6}, \tag{B.3}$$

$$\langle f f' \rangle = \langle \bar{f} \bar{f}' \rangle = 0, \qquad \langle f f'' \rangle = \langle f'' \bar{f} \rangle = -\frac{1}{6}. \tag{B.4}$$

All of the mixed averages receive a correction depending on this phase,

$$\langle f' \bar{f}' \rangle = \frac{\cos \alpha}{6}, \quad \langle f \bar{f}'' \rangle = \langle f'' \bar{f} \rangle = -\frac{\cos \alpha}{6}, \quad \langle f' \bar{f}'' \rangle = -\langle f'' \bar{f}' \rangle = -\frac{\sin \alpha}{8}, \tag{B.5}$$

$$\langle f \bar{f} \rangle = \frac{\cos \alpha}{2}, \quad \langle f'' \bar{f}'' \rangle = \frac{\cos \alpha}{10}, \quad \langle f \bar{f}' \rangle = -\langle f' \bar{f} \rangle = -\frac{\sin \alpha}{4}. \tag{B.6}$$

Consistently, $\alpha \to 0$ reduces back to the averages discussed in the main text.

Although many of the averages are modified, the resulting averaged metric and Ricci curvature do not differ much compared to (25) and (26),

$$\langle g^{(\alpha)} \rangle = \begin{pmatrix} \left(-1 + \frac{\epsilon\bar{\epsilon}}{2}\cos\alpha\right)\frac{1}{z^2} + \frac{\epsilon\bar{\epsilon}}{12}\cos\alpha & 0 & \frac{\bar{\epsilon}^2 - \epsilon^2}{24} \\ 0 & \frac{1}{z^2} & 0 \\ \frac{\bar{\epsilon}^2 - \epsilon^2}{24} & 0 & \frac{1}{z^2} + \frac{\epsilon^2 + \bar{\epsilon}^2}{24} - \frac{\epsilon\bar{\epsilon}}{40}z^2\cos\alpha \end{pmatrix}, \tag{B.7}$$

$$\text{and} \quad R^{(\alpha)} = -6 + \frac{\epsilon^2 - 2\epsilon\bar{\epsilon}\cos\alpha + \bar{\epsilon}^2}{12}z^2 + \frac{\epsilon\bar{\epsilon}}{10}z^4\cos\alpha. \tag{B.8}$$

As expected from the changes in the averages given above, the phase shift only appears in terms $\propto \epsilon\bar{\epsilon}$. A particular interesting value might be $\alpha = \frac{\pi}{2}$: in this case, the $\mathcal{O}(z^2)$ term in $\langle g^{(\alpha)}_{\phi\phi} \rangle$ vanishes and $\langle g^{(\alpha)}_{tt} \rangle = -\frac{1}{z^2}$. However, the curvature still contains a divergent piece $\propto z^2$. To summarise this brief analysis, including a relative phase shift does not yield an obvious advantage compared to our analysis in the main text.

## C  Fourth-order analysis of the disorder

Here, we explain the fourth-order expansion of the metric components and the resulting Ricci curvature and trace anomaly equation.

Expanding the metric components to fourth order in $\epsilon, \bar{\epsilon}$ yields

$$g_{tt} = (-1 + (\bar{\epsilon} - \epsilon)f + \epsilon\bar{\epsilon}f^2)\frac{1}{z^2} + \frac{\epsilon\bar{\epsilon}}{2}f'^2, \tag{C.1a}$$

$$g_{tz} = \frac{(\epsilon + \bar{\epsilon})f'}{2z}, \tag{C.1b}$$

$$g_{t\phi} = \frac{\epsilon + \bar{\epsilon}}{2}f\frac{1}{z^2} + \frac{\bar{\epsilon}^2 - \epsilon^2}{4}f'^2 + \frac{\epsilon + \bar{\epsilon}}{4}f'' + \frac{\epsilon^3 + \bar{\epsilon}^3}{4}ff'^2 - \frac{\epsilon^4 - \bar{\epsilon}^4}{4}f^2f'^2$$
$$+ \left(\frac{\epsilon^2\bar{\epsilon} + \epsilon\bar{\epsilon}^2}{16}f'^2f'' - \frac{\epsilon^3\bar{\epsilon} - \epsilon\bar{\epsilon}^3}{16}ff'^2f''\right)z^2, \tag{C.1c}$$

$$g_{zz} = \frac{1}{z^2}, \tag{C.1d}$$

$$g_{z\phi} = 0, \tag{C.1e}$$

$$
\begin{aligned}
g_{\phi\phi} &= \frac{1}{z^2} + \frac{\epsilon - \bar{\epsilon}}{2}f'' - \frac{\epsilon^2 + \bar{\epsilon}^2}{4}f'^2 - \frac{\epsilon^2 + \bar{\epsilon}^2}{2}ff'' + \frac{\epsilon^3 - \bar{\epsilon}^3}{2}\left(ff'^2 + f^2f''\right) \\
&\quad - \frac{\epsilon^4 + \bar{\epsilon}^4}{4}\left(3f^2f'^2 + f^3f''\right) + \left(-\frac{\epsilon\bar{\epsilon}}{4}f''^2 + \frac{\epsilon^2\bar{\epsilon} - \epsilon\bar{\epsilon}^2}{8}\left(f'^2f'' + 2ff''^2\right)\right. \\
&\quad + \left.\frac{\epsilon^2\bar{\epsilon}^2}{16}f'^4 - \frac{\epsilon^3\bar{\epsilon} - \epsilon^2\bar{\epsilon}^2 + \epsilon\bar{\epsilon}^3}{4}\left(ff'^2f'' + f^2f''^2\right)\right)z^2. 
\end{aligned}
\tag{C.1f}
$$

Upon averaging, the terms linear and cubic in the product of $\epsilon$ and $\bar{\epsilon}$ vanish. Using the averages

$$\langle f^4\rangle = \frac{3}{4}, \quad \langle f'^4\rangle = \frac{1}{12}, \quad \langle f^2f'^2\rangle = \frac{1}{12}, \tag{C.2}$$

$$\langle f^3f''\rangle = -\frac{1}{4}, \quad \langle ff'^2f''\rangle = -\frac{1}{36}, \quad \langle f^2f''^2\rangle = \frac{19}{180}, \tag{C.3}$$

the averaged metric components to fourth order in the disorder strengths are given by

$$\langle g_{tt}\rangle = \left(-1 + \frac{\epsilon\bar{\epsilon}}{2}\right)\frac{1}{z^2} + \frac{\epsilon\bar{\epsilon}}{12} - \frac{\epsilon^2\bar{\epsilon}^2}{192}z^2, \tag{C.4a}$$

$$\langle g_{t\phi}\rangle = \frac{\bar{\epsilon}^2 - \epsilon^2}{24} + \frac{\bar{\epsilon}^4 - \epsilon^4}{48} + \frac{\epsilon^3\bar{\epsilon} - \epsilon\bar{\epsilon}^3}{576}z^2, \tag{C.4b}$$

$$\langle g_{\phi\phi}\rangle = \frac{1}{z^2} + \frac{\epsilon^2 + \bar{\epsilon}^2}{24} - \frac{\epsilon\bar{\epsilon}}{40}z^2 + \frac{\epsilon^4 + \bar{\epsilon}^4}{16} + \frac{\epsilon^2\bar{\epsilon}^2}{576}z^2, \tag{C.4c}$$

$$\langle g_{tz}\rangle = 0 = \langle g_{z\phi}\rangle, \quad \text{and} \quad \langle g_{zz}\rangle = \frac{1}{z^2}. \tag{C.4d}$$

For these metric components, we obtain the Ricci scalar

$$
\begin{aligned}
R &= -6 + \frac{(\epsilon - \bar{\epsilon})^2}{12}z^2 + \frac{\epsilon\bar{\epsilon}}{10}z^4 + \frac{3\epsilon^4 - 2\epsilon^2\bar{\epsilon}^2 + 3\bar{\epsilon}^4}{24}z^2 \\
&\quad - \frac{\epsilon^4 - 2\epsilon^3\bar{\epsilon} + 6\epsilon^2\bar{\epsilon}^2 - 2\epsilon\bar{\epsilon}^3 + \bar{\epsilon}^4}{288}z^4 - \frac{5\epsilon^4\bar{\epsilon} + 4\epsilon^2\bar{\epsilon}^2 + 5\epsilon\bar{\epsilon}^3}{480}z^6 + \frac{3\epsilon^2\bar{\epsilon}^2}{400}z^8,
\end{aligned}
\tag{C.5}
$$

and, using (30),

$$\langle\langle T_{ij}^{\text{ren}}[\langle\gamma\rangle]\rangle\rangle = \frac{c}{288\pi}\begin{pmatrix} \epsilon^2 + \bar{\epsilon}^2 + \frac{3}{2}\epsilon^4 - \frac{1}{2}\epsilon^3\bar{\epsilon} - \frac{1}{2}\epsilon\bar{\epsilon}^3 + \frac{3}{2}\bar{\epsilon}^4 & \bar{\epsilon}^2 - \epsilon^2 + \frac{\bar{\epsilon}^4 - \epsilon^4}{2} \\ \bar{\epsilon}^2 - \epsilon^2 + \frac{\bar{\epsilon}^4 - \epsilon^4}{2} & 2\epsilon\bar{\epsilon} + \epsilon^2\bar{\epsilon}^2 \end{pmatrix}. \tag{C.6}$$

The trace of the energy-momentum tensor is given by

$$\text{tr}\left(\langle\langle T_{ij}^{\text{ren}}[\langle\gamma\rangle]\rangle\rangle\right) = -\frac{c(\epsilon - \bar{\epsilon}^2)}{288\pi} - \frac{c(3\epsilon^4 - 4\epsilon^2\bar{\epsilon}^2 + 3\bar{\epsilon}^4)}{576\pi}. \tag{C.7}$$

Comparing these results to (26), (30) and (33) shows that while we of course obtain non-trivial corrections by including the fourth order, neither the IR divergence of the curvature nor the non-vanishing trace of the boundary energy-momentum tensor are resolved by this inclusion. Therefore we conclude that going to higher orders in perturbation theory only leads to quantitative changes, but qualitatively we do not learn more about the properties of the bulk curvature and the trace of the boundary energy-momentum tensor.

## D  Numerical evaluation of the Ricci scalar

Here, we give details on the numerical average evaluation. In particular, we compute the averaged metric components and the corresponding Ricci scalar without expanding in $\epsilon$. We show this explicitly for $\epsilon = \bar{\epsilon}$.

For the numerical evaluation, we use *Mathematica*. The metric components are obtained as follows. The components of the connection are typed into *Mathematica* such that they only depend on $\mu$ and $\bar{\mu}$. Using (14), each metric component can be computed. The metric is denoted as gPP. The chemical potentials are defined as

```
1    μ[ϕ_]:=1+ ε/√N Sum[Cos[n/N ϕ+γ[[n]]],{n,1,N}]
2    μ̄[ϕ_]:=-1+ ε̄/√N Sum[Cos[n/N ϕ+γ[[n]]],{n,1,N}]
```

To calculate the averaged metric components, we calculate the components for $N$ realisations and afterwards take the mean of them. To do so, we first define empty lists for each component, such as

```
1    gttR={}
```

for the $tt$-component. We then run a For loop to evaluate the metric components for particular realisations of the disorder. Within the loop, for each instance, new random phases are computed as a list of length $N$. The phases are, by their definition, automatically inserted into the chemical potentials $\mu, \bar{\mu}$. These particular realisations of $\mu$ and $\bar{\mu}$ are put into the metric components. After evaluating them, the component is included in the lists priorly defined, using AppendTo. After the loop, each of the lists is averaged by using Mean. In *Mathematica*, this looks as follows (again written only for the $tt$-component):

```
1    For[i=0,i<N,i++;{γ=Table[RandomReal[{0,2*Pi}],{k,1,N}],AppendTo[
     gttR,gPP[[1,1]]/.{μ->μ[ϕ_]}]
2    gttM=Mean[gttR]
```

We proceed in this way for every component. In practice, to ensure all components are computed with the same random phases, we run one loop containing the corresponding operations and the function AppendTo for every metric component. In this way, we obtain the averaged metric without expanding in $\epsilon, \bar{\epsilon}$. By using the standard formulae, we can calculate the Ricci scalar. This code works well at least up to $N = 15$, which is already a fairly good approximation, given that the resulting error is of the order $\frac{1}{N} \sim 10\%$. The resulting Ricci scalar shows, up to finite $N$ effects, the same diverging behaviour in the IR region as the Ricci scalar computed in the main text by expanding the metric components to second order in $\epsilon, \bar{\epsilon}$. In particular, evaluating the Ricci scalar obtained numerically for $\epsilon = \bar{\epsilon}$ and $\epsilon$ small shows a good match between the numerics and the analytical perturbative calculation, as displayed in figure 2. Comparing the two figures 2a and 2b, we also see how a larger $N$ leads to dampening of the fluctuations in $\phi$ direction.

## E  Details of the QNEC calculation

In this appendix, we explain in detail how the results for the QNEC conditions are obtained following the method discussed in section 2.6 in [49]. We will first explain the idea and give explicit expressions afterwards.

The calculation results from the following idea: the EE $S$ is computed by a static geodesic of length $l$, connecting two points in the boundary CFT. To obtain QNEC, we need to know how $S$ changes under small null deformations of the entangling region. For concreteness, this means we change one of the endpoints by a vector of the form $k = \lambda(1,1)$, where $\lambda$ is assumed to be small compared to $l$. Therefore, and also due to the fact that we will need only

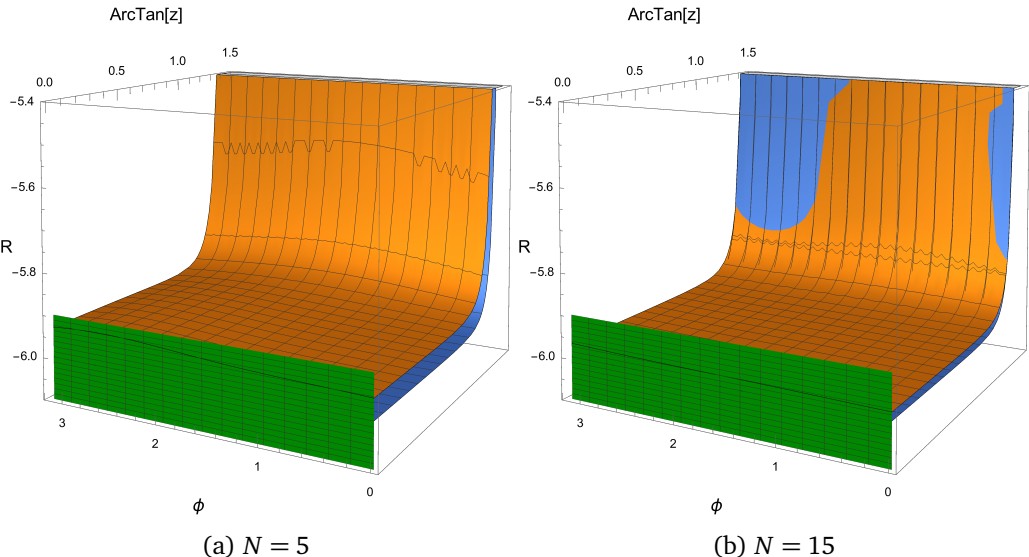

(a) $N = 5$          (b) $N = 15$

Figure 2: Displayed in both images is the Ricci scalar for the disordered Poincaré patch evaluated numerically (orange) in comparison to the analytic result (light blue). The green plane represents a $\pm\epsilon$ band. These result are obtained for equal disorder strengths $\epsilon = \bar{\epsilon} = 0.1$ and for $N = 5$ (figure 2a) and $N = 15$ (figure 2b) in the disordered chemical potentials (16), (17). For the plots, the coordinate $\theta = \arctan(z)$ is introduced. In the IR region $z \to \infty$, i.e. $\theta \to \frac{\pi}{2}$, the curvature is divergent both numerically and analytically (compare (26)). The fact that the curvature varies in $\phi$ direction is a finite $N$ effect. As can be seen by comparing the two plots, the curvature variations in figure (2b) are damped due to the larger value of $N$.

second and first derivatives of $S$ evaluated at $\lambda = 0$, we can always work perturbatively in $\lambda$. For our disordered metric, we will also work perturbatively in $\epsilon$. The calculation works as follows: we choose the spatial boundary coordinate $\phi$ as affine parameter to denote the geodesic Lagrangian $\mathcal{L}$ in terms of $\dot{t}$, $\dot{z}$ and $z$, where a dot refers to a derivative w.r.t $\phi$. This makes use of the existence of the Killing vectors $\partial_t$ and $\partial_\phi$, i.e. the metric components depend only on the radial coordinate $z$. The area, and thereby the EE, is then given by the integral of $\mathcal{L}$ over $\phi$. It is, however, more convenient to reexpress this as an integral over $z$. To do so, we make use of the Noether charges of $\partial_\phi$ and $\partial_t$, $Q_1$ and $Q_2$ respectively, where for calculational convenience we evaluate $Q_1$ at the turning point $z_*$ of the geodesic. It is then possible to express $\dot{t}$ and $\dot{z}$ as functions $h_t$ and $h_z$ depending on $z$, $z_*$ and $\Lambda$, where $\Lambda$ is a combination of the two charges. For example, one may take $\Lambda = Q_2/Q_1$, which turns out to be very convenient in the case of diagonal metrics. Furthermore, $\Lambda$ has to vanish when the null deformation is turned off. Therefore, $\Lambda$ and $\lambda$ are linearly related to lowest order, such that expansions in $\Lambda$ are also possible. The null deformation changes the extent of the entangling region by $\lambda$ both in the spatial and temporal direction. However, using $h_t$ and $h_z$, the change of size of the entangling interval can be calculated in terms of $z_*$ and $\Lambda$, such that these two parameters can be expressed in terms of $l$ and $\lambda$. Finally, to calculate the EE, the integral can be written as an integral over $z$, using again $h_z$, where $z_*$ and $\Lambda$ are expressed in terms of $l$ and $\lambda$. Having obtained this result, it is straightforward to calculate the QNEC combination in (35).

After this more general discussion, we present in detail the calculation for the metric in (25). The geodesic Lagrangian

$$\mathcal{L}(\dot{t}, \dot{z}, z) = \sqrt{\langle g_{\phi\phi} \rangle + 2 \langle g_{t\phi} \rangle \dot{t} + \langle g_{tt} \rangle \dot{t}^2 + \frac{\dot{z}^2}{z^2}}, \tag{E.1}$$

has the Noether charges

$$Q_1 = \dot{z}\frac{\partial \mathcal{L}}{\partial \dot{z}} + \dot{t}\frac{\partial \mathcal{L}}{\partial \dot{t}} - \mathcal{L} = -\frac{\langle g_{\phi\phi}\rangle + \langle g_{t\phi}\rangle \dot{t}}{\mathcal{L}}, \tag{E.2}$$

$$Q_2 = \frac{\partial \mathcal{L}}{\partial \dot{t}} = \frac{\langle g_{t\phi}\rangle + \langle g_{tt}\rangle \dot{t}}{\mathcal{L}}. \tag{E.3}$$

Evaluating $Q_1$ at $z_*$ and solving for $\dot{t}$ and $\dot{z}$, the solutions are of the form

$$\dot{t} = \Lambda h_t(z, z_*, \Lambda), \quad \text{and} \quad \dot{z} = h_z(z, z_*, \Lambda). \tag{E.4}$$

Inserting the metric components (25), we find to $\mathcal{O}(\epsilon^2)$

$$h_t = \Lambda\left[1 + \frac{z^2\epsilon^2}{24\Lambda}\left(-(1-\Lambda+\Lambda^2) + \left(2-\frac{3}{5}z^2\right)a\Lambda + (1+\Lambda+\Lambda^2)a^2\right)\right], \tag{E.5}$$

$$\begin{aligned}
h_z = &-\frac{\sqrt{(z_*^2-z^2)(1-\Lambda^2)}}{z} - \frac{\epsilon^2}{240z}\sqrt{\frac{z_*^2-z^2}{1-\Lambda^2}}\Big[5(2\Lambda-1)z_*^2 + 5(1-2\Lambda^2+2\Lambda^3)z^2 \\
&+ \left(3z_*^4 - 10z_*^2\Lambda^2 + z^2(3z_*^2 - 10\Lambda^2) + z^4(6\Lambda^2-3)\right)a \\
&- \left(5(1+2\Lambda)z_*^2 + 5(-1+2\Lambda^2+2\Lambda^3)z^2\right)a^2\Big],
\end{aligned} \tag{E.6}$$

where $\Lambda = \frac{Q_2}{Q_1}$ and $a = \frac{\bar{\epsilon}}{\epsilon}$ is the ratio of the disorder strengths.

The next step is to calculate the expressions for the spatial and temporal deformation of the entangling interval. In principle, they are given by $\phi$- and $t$-integrals respectively, but using the above functions, both can be computed as a $z$-integral:

$$\lambda = \int_0^\lambda \mathrm{d}t = 2\Lambda \int_{z_*}^0 \mathrm{d}z\,\frac{h_t}{h_z}, \tag{E.7}$$

$$l + \lambda = \int_0^{l+\lambda} \mathrm{d}\phi = 2\int_{z_*}^0 \frac{\mathrm{d}z}{h_z}. \tag{E.8}$$

These integrals can be performed after expanding the argument to second order in $\epsilon$. Solving for $\Lambda$ and $z_*$ yields

$$\Lambda = \frac{\lambda}{2z_*} + \epsilon^2\left[\frac{a^2-1}{36}z_*^2 + \frac{-125(1+a^2)+(99z_*^2-200)a}{7200}z_*\lambda + \frac{1-a^2}{96}\lambda^2\right], \tag{E.9}$$

$$\begin{aligned}
z_* = &\frac{l}{2}\left(1 + \frac{51al^2 - 100(1+a^2)}{57600}l^2\epsilon^2\right) + \frac{\lambda}{2}\left(1 + \frac{51al^2 - 20 - 100a^2}{11520}l^2\epsilon^2\right) \\
&+ \frac{\lambda^2}{4l}\left(1 - \frac{300 - 1300a^2 - (400-867l^2)a}{57600}l^2\epsilon^2\right).
\end{aligned} \tag{E.10}$$

Finally, the area integral can be evaluated to give a result depending only on $l$, $\lambda$ and the radial cutoff $z_{\mathrm{cut}}$

$$A = 2\int_{z_*}^{z_{\mathrm{cut}}} \mathrm{d}z\,\frac{\mathcal{L}(\Lambda h_t, h_z, z)}{h_z}. \tag{E.11}$$

The explicit result using $l \gg z_{\text{cut}}$, expanded to second order in $\epsilon$ and $\lambda$, is given by

$$S(\lambda) = \frac{c}{3} \ln \frac{l}{z_{\text{cut}}} + \frac{c\lambda}{3l} - \frac{c\lambda^2}{3l^2} + \frac{c\epsilon^2}{864} \left[ (1+a^2)l^2 - \frac{3al^4}{25} + \left( 4a^2 - \frac{12al^2}{25} \right) l\lambda \right.$$
$$\left. + \left( -1 + 2a + 3a^2 - \frac{3al^2}{5} \right) \lambda^2 \right]. \qquad \text{(E.12)}$$

Taking derivatives w.r.t $\lambda$ and setting $\lambda = 0$ afterwards, we find the result given in (38). While the above $S$ is calculated by making use of $l \gg z_{\text{cut}}$ in the last step, we find the same result (38) when using the full result from (E.11), without assuming large $l$. The calculation for the resummed geometry (45) works analogously and is even considerably shorter.

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
