# Peer review of "Disorder in AdS$_3$/CFT$_2$"

_SciPost Physics, doi:SciPost Phys. 16, 017 (2024)_

## Round 2 · Referee Report · Anonymous (Referee 1) · 2022-5-29

Report

This paper deals with a challenging and interesting topic, that of strongly coupled disordered systems, in a rather new setup. They study disorder via the AdS3/CFT2 duality by means of the Chern-Simons (CS) formulation of AdS gravity. The manuscript is clearly written and the computations and results are nicely described and presented.

Although this work presents an interesting approach to the treatment of disorder via holography; in its present form I am not convinced that the manuscript meets any of the four 'Expectations' acceptance criteria of the journal. I shall go through the article and raise questions related to the points I believe are the weak or unclear steps of this work.

The paper starts with a nice introduction where the authors clearly set the problem and the progress made in past studies of holographic disorder. Then in section 2 they introduce the CS formulation of 3d gravity and how disorder will be introduced through the chemical potentials of the two gauge fields. They show how in CS gravity one can easily obtain the resulting metric for a chosen chemical potential. And here comes my first question:
[from here on the numbering of the questions is related to the corresponding section in the paper]

2.1) If I understood the computations correctly they can in principle obtain an expression of the metric in terms of the disordered chemical potentials (16 and 17) and their derivatives (and possibly ratios of mu, \bar mu and their derivatives). Do I correctly understand that they truncate the solution at 2nd order in the disorder strength only to be able to analytically perform the disorder average?
[If an expression of the metric non-perturbative in disorder is at hand maybe many of the points in the paper could be made stronger by a couple of simulations]

2.2) In the expression for the metric in the appendix (eq. 62) there appear IR divergent terms (~z^2) in g_{phi phi}. Am I right to assume that the expressions in eq 62 are correct in z but truncated at quadratic order in \epsilon?
Then one could worry that unless somehow the z^2 terms vanish upon going to higher orders in epsilon (or beyond perturbative disorder) this is already indicating that disorder is likely to give rise to a divergent IR (something that would not be surprising if the disorder is marginally relevant). Going back to the comment 2.1) above: it would probably be quite easy to get a numerical estimate of the fate of these z^2 terms beyond the perturbative expansion.

3.1) Section 3 is devoted to the study of the state dual to the averaged metric. I would like to ask first about the state resulting from averaging over realisations. The authors mention that the resulting energy-momentum tensor satisfies the QNEC (as opposed to the one resulting from the average metric). I would like to ask about the energy momentum tensor resulting from averaging over realisations and how it compares to the energy momentum tensor they obtain in section 4 after the 'resummation' procedure. Does it also imply the introduction of mass and angular momentum?

3.2) Small question about sections 3.1 and 3.2: given that the average metric does not solve the Einstein equations, is it surprising that the standard results for the trace anomaly and the QNEC do not hold?

In section 4 the authors set on to finding a 'resummed' averaged metric

4.1) They define the procedure as resummation and refer to previous holographic examples. In those, a non-divergent averaged metric was found after first modifying the metric ansatz and then solving the EoMs for the 'resummed' metric ansatz. This seems out of reach within the CS formalism used by the authors. And in my view this 'resummation' is a weakness of the paper since one could argue that the 'resummed' averaged metric they obtain in section 4 is just a new solution of vacuum Einstein's equations whose connection to the initial disorder problem seems tenuous.

4.2) A small question about the 'resummed' averaged metric. Its g_{tt} basically reads g_{tt} = -1/z^2 +1/24 (\epsilon^2 +\bar\epsilon^2), right?
What should I make of the value of z where this g_{tt} vanishes?

4.2) In the 'resummation' process the authors modify the averaged metric in a somewhat minimalistic way such that they arrive to a metric that now solves the vacuum Einstein equations and thus verifies the trace anomaly and QNEC conditions. Ideally it should be made much more clear why one should expect that this modified metric is actually describing the disordered system (for small disorder). Since this modified metric does not follow from solving the initial problem of introducing disorder via the CS chemical potentials I think a stronger case needs to be done about this. (A possible way to go is to just obtain 'numerical' solutions if, as I asked in 2.1, the expression for the non-truncated metric in terms of the disorder distribution is at hand).
I am also worried about the lack of justification for some of the choices in the 'resummation', for instance at some point some coefficients are set to zero 'in order to introduce as little change by the 'resummation' parameters as possible'. Since by this process one is already obtaining a quite different metric from the one we started with it is not obvious to me that this justification holds.

All in all, I think this work presents a nice idea but, up to my understanding of it, it needs to be improved in some key points to meet any of the 'Expectations' criteria of this journal. As I hope is clear from my comments&questions above, I believe that the results presented in the manuscript do not clearly show that the proposed 'resummed metric' is actually describing the effect of the introduction of disorder via the CS chemical potentials as is the stated aim of the work.
Maybe the connection between the 'resummed' metric and the actual effect of the disordered chemical potentials can be supported by further analytical computations (via the Einstein equations?) or a couple of numerical simulations that should be easy to perform (just doing some averages over expressions containing maybe complicated combinations of the disordered distribution) if the authors have an analytic expression for the metric in terms of the disorder distribution. Alternatively, maybe the authors could consider aiming for publication in SciPost Physics core which has less stringent acceptance criteria in terms of exceptionality.

---

## Round 2 · Referee Report · Anonymous (Referee 2) · 2022-6-2

Report

The authors discuss a holographic model corresponding to a 2-dimensional CFT deformed by marginally relevant quenched disorder and in particular they study the effects of disorder on the averaged geometry. Disorder is introduced in a novel way by using a Chern-Simons formulation of AdS3 gravity. The attempt is interesting and the results deserve publication in some form. Nevertheless, I agree with the other anonymous Referee that some technical parts of the paper must be clarified and the analytic results should be supported by some additional numerical confirmations.

My main complaint regards the role/meaning of the unphysical features mentioned in the manuscript and the validity of the Poincare-Lindstedt inspired resummation. More precisely:

  1. In the text, I found different statements regarding the nature of these unphysical features. In the abstract the authors state that these unphysical features are due to the disorder average. In pages 3 and 15, they say that these features are a signal of the breakdown of the perturbative expansion. Also, in page 15 they suggest the possibility that these features are actually a signature of the marginal relevance of the disorder deformation. I believe that a clarification on these lines is necessary and important. Are these features really unphysical? Or are they telling us something about the nature of the introduced disorder and/or the techniques used?

  2. The most important point is probably the question whether the resummation commutes or not with the disorder average. In the References cited where this resummation is employed, the resummation is operatively performed before the average. On the contrary, the authors here introduce the resummation after the average. At page 15, they claim that the two commute because the functions used to regularize the metric (after the average) do not depend on the random phases. This statement is highly not obvious to me. First, after the average, no phases can appear in any case in the regularization functions, so I am not sure how this can be an argument for the commutation of the two operations. Second, if really this was the case, one could take the same phases independent terms and regularize the metric before the average, i.e., commuting the two operations. Then, the average over the regularization terms would be trivial since they do not explicitly depend on the phases. The Authors should provide more solid arguments for this commutation and perhaps confirm it with some numerical computations as suggested by the other Referee.

(3) The resummation not only fixes the ~z^2 unwanted terms in the metric, curvature, etc but it seems also to modify finite physical observables. For example (32) and (56) seem different. How can one understand this? In my opinion, some comments on this point are in order.

Finally, some minor comments.

  1. In figure 1, the light blue line separating the region could be probably removed.

(5)The introduced disorder clearly breaks the translational symmetry of the dual field theory. From symmetry arguments, one would therefore expect the corresponding gravity theory to display a mass for the graviton, probably modifying the Einstein equations of motion (see for example PhysRevLett.112.071602). This is somehow connected with some brief discussion in the conclusions about topological massive gravity. Could this be linked to the fact that the average metric does not solve anymore the Einstein equations in vacuum? Is this mass appearing somewhere in your setup or do you believe that it would appear only at the level of the fluctuations ?

(6)The Authors introduce the sources of disorder in the gauge fields. Is it possible to show to which components of the metric they correspond to (using Eq.(14) for example)? That would help to gain some physical intuition behind the technical construction.

  1. Some additional references about disorder in holography and CFTs should be added in the introductory part. E.g. : 1509.02547, 1401.7993, 1505.05171, 1802.08650,1409.6875, ...

  2. More importantly, I am quite surprised that 2004.06543, 2110.11978 are not mentioned. Not only they contain important developments following other References mentioned in the introduction but they could be even related to the IR divergences seen by the Authors. I suggest to add them and comment on these.

  3. Finally, when the Authors mention the physics of Anderson localization (which seems to be the long-range task of this exploration somehow) they should provide a few more details about it and in particular about the existing results in the holographic literature. Ref.[21] for example made an important statement about the possibility of (not) having Anderson localization in holography (or at least in simple holographic models) [btw. 1511.05970 could be also mentioned]. Later on, some counterexamples of those statements appear by adding extra bulk couplings (1601.07897,1602.01067 ). Finally, 1711.10953 provided an interesting view on the topic. In my opinion, that part could be a bit enlarged. Also, can Anderson localization appear in a classical theory or is it necessarily tight to quantum effects (interference)? If quantum effects are necessary, then it is at least disputable whether large N holographic models could provide any help in this direction. A comment about this point would help the Reader.

---

## Round 4 · Referee Report · Anonymous · 2022-11-14

Report

The Authors have seriously considered all the comments in my previous report. I am satisfied with their reply and with the effort put in the revision. I am therefore happy to recommend the publication of the manuscript in its actual form.

---

## Round 4 · Referee Report · Anonymous · 2022-11-29

Report

I have to start by appologising for the delayed report. Next, let me hank the authors for the effort put in addressing my comments.

Since my main concern regarding this work was that of the relation between the 'resummed' solution and the disordered chemical potential, I shall ask directly about this point.

1)It is very nice that the authors have been able to numerically compute the disordered metric non-perturbatively. I have only one question about this computation. It seems very surprising that they cannot compute the Ricci for metrics resulting from averaging over more than 15 modes. Do I understand correctly that the result of averaging over realisations is a metric that depends on both phi and z? Then the Ricci tensor of this metric is computed numerically, correct? Why is the numerical computation of Ricci more difficult when the metric results from averaging over say 100 realisations than when it results from averaging over 15 realisations?
I am sure I am missing something cause I would expect any numerical package to be able to easily compute Ricci for a given metric (defined in terms of numerical functions -as opposed to analytical expressions).

2) As the authors say, the numerical analysis seems to be telling us that the divergence found towards the IR in the perturbative approach is truly there for the all order analysis. This invites the obvious question about the physical relationship between the regular 'resummed' geometry and the physical problem of the geometry induced by the disordered chemical potentials. One would expect that, as it happened in e.g. ref [14], the resummed perturbative geometry agrees with the numerical solution for small enough value of the disorder strength. And if I am not wrong this is not happening here since while the numerical solution is singular in the IR, the 'resummed' one is not.

2.1) Related to this point I find it interesting that the authors are able to arrive to the 'resummed' metric from solving the vacuum Einstein equations. I would like to ask if it is obvious that it is the vacuum equations the ones that matter for this case (in view of the results for the dual energy momentum tensor could one expect the solution to result from a gravity plus matter setup?).

3.1) A less pressing matter I mentioned in my initial report (3.1) was the question about the energy momentum tensor averaged over realizations. If I understood correctly the tensor for each realization is healthy and satisfies QNEC, so I was wondering in that question how the tensor looked like if one averages over realizations (i.e. compute $T_{\mu\nu}$ for each realization and then average), and how it compared to the tensor resulting from the averaged metric.

In summary, I think the apparent disagreement between the numerical all orders result (singular geometry) and the 'resummed' regular metric needs to be clarified further. Is there an obvious reason why the numerical procedure should not result in the regular resummed geometry for small enough disorder strength? (I apologise if I have missed an obvious reason why the resummed perturbative solution and the numerical one should not agree at all).

---

## Round 4 · Author Response

Dear Editor,
we thank both the referees for carefully reading our manuscript and for the detailed comments that we address below. A detailed list of all changes is given below. First however, given that we have answered all of the referees questions in detail, we kindly ask to reconsider our submission for publication in SciPost Physics.
We first address the comments of report 1 dated 29.05.2022 and afterwards the comments of report 2 dated 02.06.2022.

*** report dated to 29.05.2022

2.1) Indeed, the metric can be obtained without involving perturbation theory from the gauge fields by using eq.~(14) in our paper. The resulting components are functions of $\mu$, $\bar{\mu}$, and their derivatives. For clarity, we have now included the non-perturbative expressions for the metric components in Appendix A, eq.~(66). The charges $\cal L$ and $\bar{\cal L}$ appearing in these equations are given by eq.~(12) and eq.~(13), respectively. Averaging these metric components analytically involves integrals for which closed expressions could not be found. Therefore, as the referee expected, we expand in $\epsilon$ and $\bar{\epsilon}$ to be able to analytically compute the averages. We now state this more clearly below eq.~(21). Regarding the suggestion about using numerics, we refer to the next question where we made use of numerical analysis to solidify some of our statements.

2.2) The metric components given in eq.~(67) [former eq.~(62)] are expanded in the disorder strengths $\epsilon,\bar{\epsilon}$, not in the radial coordinate $z$. Going to higher orders in $\epsilon$ does not lead to a qualitative change of the $z^2$ terms, and hence the infrared divergence persists. We confirm this now, once by calculating the average of the fourth-order expansion of the metric components (eq.~(79)) in Appendix C and once by a numerical analysis described in Appendix D. In both cases, the IR divergence can be found, see eq.~80 and fig.~2. Hence, we confirm the referee's expectation that the marginally relevant disorder gives rise to this IR singularity.

3.1) Before addressing the actual questions, we clarify a potential misunderstanding. The energy-momentum tensor eq.~(32) resulting from the averaged metric does not satisfy QNEC for all values of the disorder strengths $\epsilon$ and $\bar{\epsilon}$. We analysed this in sec.~3.2, cf.~Fig.~1 in the main text. However, as discussed in sec.~4, after resummation, the resulting energy-momentum tensor fulfills QNEC. We furthermore point out that the fulfillment of QNEC after resummation is not a requirement for fixing the constants in the resummation procedure, but directly follows from it, cf.~eqs.~(61) and (62).

Both the energy-momentum tensors in eq.~(32) (before resummation) and in eqs.~(47)/(56) imply that the disorder sources mass and angular momentum. We state this below eq.~(32) for the energy momentum-tensor before resummation.

The two energy-momentum tensors compare in the following way. The state in eq.~(32) does not satisfy the trace anomaly equation relating the trace to the boundary Ricci scalar. In addition, it does not satisfy QNEC in general. Its dual bulk geometry contains curvature singularities in the IR region (eq.~(26)), is not an AdS spacetime everywhere (eq.~(27)), and has a singularity in the causal structure (eq.~(28)). All of these properties are cured by the resummation procedure performed in sec. 4.

3.2) We agree that in hindsight, the result for the trace anomaly might not be surprising. However, to the best of our knowledge, it is not a priori clear that for a metric not solving the vacuum Einstein equations, the trace anomaly equation must be violated. A counterexample of a non-vacuum solution where the trace anomaly equation holds can be found e.g. in ``Marc Henneaux, Cristian Martinez, Ricardo Troncoso, and Jorge Zanelli.
Black holes and asymptotics of 2+1 gravity coupled to a scalar field.
Phys. Rev. D, 65:104007, 2002''.

Regarding QNEC, there is no direct relation between a metric not satisfying the vacuum Einstein equations and the possible violation of QNEC. Rather, there exist counter examples which can be found in reference [49] in the main text and references therein. Generically, the QNEC inequality may not be saturated in a state with bulk matter switched on, but that does not mean it has to be violated.

In order to further analyse the origin of violation of the trace anomaly equation and QNEC, we would need knowledge about the effective theory to which our averaged metric is a solution to. Such an effective theory is, at the time of writing, however not available to us; we comment on this in more detail in the conclusion section of the main text. The results for the trace anomaly and QNEC as we explain them in the paper provide help in constraining the search for an effective theory.

4.1) To define our resummation, we proceed in a similar way as the references pointed out in the text. We modify the metric and demand the resummation to cure the divergent behaviour. The fact that we do so in eq.~(41) after averaging is only due to convenience in the calculation. Since, by definition, the resummation functions only depend on $z$, and in particular not on the random phases $\gamma_n$, the modification commutes with the averaging procedure. Concerning the relation to the CS formulation, so far, staying in radial gauge, we could not find a good ansatz for the resummation on the level of the Chern--Simons (CS) fields. However, since on the classical level the Einstein--Hilbert (EH) formulation is equivalent to the CS formulation, there has to be a way to implement the resummation also in the CS formalism, most probably by going beyond radial gauge. As we comment in the conclusion, we leave this task for future work.

Furthermore, there is a related question at which level the average is to be imposed. In our work, we choose to impose it after the metric components are obtained. It might also be interesting to directly average the CS connection. Since the metric is bilinear in the connection, the result will be different, implying that the classical equivalence of EH and CS interferes with introducing disorder. We have added some comments regarding this issue below eq.~(20).

Regarding the relation between the resummed metric and the one originally obtained from the disorder, we point out that the resummation is not performed arbitrarily. Throughout the resummation process described in sec.~4, various reasonable physical conditions are imposed on the metric. These conditions lead to constraints on the resummation parameters fixing the parameters uniquely, up to freedom of choice for $b_{12},b_{22},b_{32}$, whose choice does not violate QNEC. We discuss the determination of $b_{12},b_{22},b_{32}$ in the last step of the resummation explained on p.13-15 in the main text.

To say more about the relation of the resummed and the averaged metric, we inserted the resummed metric with general open coefficients eq. (41) in the Einstein equations. Expanding to second order in $\epsilon,\bar{\epsilon}$, where we assume again that the open coefficients can be thought of as a power series in $\epsilon,\bar{\epsilon}$ and demanding that the Einstein equations are satisfied yields differential equations fixing the $z$ dependence of the coefficients $\alpha(z)$ and $\beta(z)$. Up to an integration constant, solving the differential equations directly yields the resummed solution eq. (45) with $b_{12}=b_{22}=b_{32}=0$. The integration constant then again has to be set to zero using the BPS condition as described in section 4. So we find that the resummed metric is not a generic new solution to the Einstein equations, but is fixed by solving differential equations following from the averaged metric including the resummation parameters. We have added these explanations in the main text after discussing the BPS bound at the bottom of page 14 and the top of page 15, in the paragraph containing the new equations (57)-(60).

4.2a) We thank the referee for asking this question. Indeed, the position $z_h$ where $g_{tt}(z_h)=0$ may be interpreted as a horizon. Denoting the induced mass and angular momentum as $M=\frac{\epsilon^2+\bar{\epsilon}^2}{24}$ and $J=\frac{\bar{\epsilon}^2-\epsilon^2}{24}$, the $tt$-component of the resummed metric may suggestively be written as
\begin{align}
\d s^2=-\frac{1}{z^2}+M+\mathcal{O}(\epsilon^4)\stackrel{?}{=}-\frac{1}{z^2}+M+\frac{M^2-J^2}{4}z^2.
\end{align}
To see if this really is the case, the fourth-order analysis now included in Appendix C will have to undergo the resummation procedure. Accompanying this, an effective description in the Chern--Simons formulation has to reproduce these fourth-order considerations using an appropriate effective gauge field. In order for this to describe a black hole solution, the gauge field has to satisfy the holonomy condition. We intend to work this out in more detail to see whether disorder can source black hole solutions.

4.2b) The resummation procedure we perform yields a reasonable physical system connected to the disorder we initially introduce. This is now made more clear also by a slightly different resummation procedure that we describe in the paragraph around the new equations (57)-(60), namely by fixing the resummation functions via solving differential equations resulting from the Einstein equations. As we now show by the analysis in Appendix C, going to fourth-order in perturbation theory does neither resolve the curvature divergence in the IR region nor the violation of the trace anomaly equation. Moreover, in Appendix D we show numerically that the curvature divergence is also present non-perturbatively.

We set these resummation parameters to zero in order to minimise the influence of the resummation on the result. In more detail, the parameters receive bounds by the BPS analysis performed in section 4. By these bounds, we determine the minimal amount of extra mass that we have to include via the resummation procedure in order to obtain a geometry satisfying the BPS bound. In particular, the minimal amount we have to include yields a resummed metric saturating the bound for generic $\epsilon$, $\bar{\epsilon}$. Since we do not want to include more mass by the resummation as necessary, we set the corresponding parameters to the lowest possible value. Following the above lines, we have enlarged the discussion about this point in the paragraph between eqs. (52) and (53) on page 14.

*** report dated to 02.06.2022

(1) The marginal relevance of our choice of disorder implies that we should expect drastic changes in the IR bulk region. Therefore, as we point out in the paragraph below eq. (29), we are not surprised to encounter divergent quantities. In the infrared, where the divergent behaviour appears, the perturbative treatment breaks down since the perturbative corrections grow unbounded and eventually become much larger than the zeroth order of the quantity in question. Therefore, the marginal relevance of our disorder leads to a breakdown of the perturbative treatment in certain limits.

In our approach, we make use of the fact that the three-dimensional equations of motion of gravity in a negatively curved spacetime can be solved exactly. This yields, for every choice of (static) chemical potentials $\mu(\phi),\bar{\mu}(\phi)$, the solutions for the charges ${\cal L},\bar{\cal L}$ (eq. (12) and eq. (13) in the main text). For every choice of the chemical potentials, which in particular includes our disordered chemical potentials (eq. (16) and eq. (17) in the main text), these charges are fixed such that the spacetime has constant negative curvature, i.e. is locally AdS$_3$. However, averaging the metric components results in a geometry which, to $\mathcal{O}(\epsilon^2)$, does not have constant negative curvature. In particular, the curvature diverges for $z\to\infty$ (see eq. (26) in the main test). Furthermore, this geometry leads to an expression for the entanglement entropy that can be negative for sufficiently large length $l$ of the RT geodesic (see eq. (40) in the main text). Hence we conclude that these features, which are clearly unphysical, result operationally from the disorder average.

To summarise, we see that using marginally relevant disorder leads to a breakdown of perturbation theory in certain limits. In our approach, this breakdown of perturbation theory is indicated by quantities showing unphysical behaviour. This unphysical behaviour is due to computing quantities from the disorder averaged metric without the resummation being implemented. Along these lines, we have enlarged the explanations in the second paragraph on p. 4 and the first and second paragraphs on p. 16 to make the connection between the unphysical behaviour, the breakdown of perturbation theory, and the marginal relevance, more clear. Moreover, we added clarifying comments below eq. (24), in the paragraph below eq. (29) and eq. (40).

(2) The resummation functions, as written in eq.~(42) in the main text, are defined to only depend on the radial coordinate $z$. This is in line with the resummation functions of the references cited; there as well, the resummation functions depended only on the radial coordinates and in particular not on the random phases. Therefore, the resummation functions commute with the averaging procedure eq.~(18). More generally, due to the definition in eq.~(18), for any function $f$ that does not depend on the random phases it holds that
\begin{align}
\expval{f}=\lim\limits_{N\to\infty}\int_0^{2\pi}\prod_{n=1}^N\frac{\d\gamma_n}{2\pi}\,f=f\lim\limits_{N\to\infty}1=f.
\end{align}
In this sense, we confirm the statement of the referee that the average over the resummation functions is trivial. For the specific case of the metric components, the commutation can be seen as follows. Consider resumming the metric components before disorder, that is
\begin{align}
g_{tt}\to\frac{g_{tt}}{\alpha(z)}\quad\text{and}\quad g_{\phi\phi}\to\frac{g_{\phi\phi}}{\beta(z)}.
\end{align}
The average of the resummed metric components is then given by
\begin{align}
\expval{\frac{g_{tt}}{\alpha(z)}}=\frac{\expval{g_{tt}}}{\alpha(z)},\quad\expval{\frac{g_{\phi\phi}}{\beta(z)}}=\frac{\expval{g_{\phi\phi}}}{\beta(z)}.
\end{align}
For an appropriate choice of the coefficients within $\alpha$ and $\beta$, the resummed geometry is well-behaved. From the above two equations it is clear that the resummation commutes with the average. In the main text, in line with the approaches of the cited references using a resummation, we are ultimately interested in the averaged geometry. In particular, we do not demand that the geometry should be regular in each disorder realisation, so before performing the average. Therefore, we are free to include the resummation functions before or after averaging. Following this reasoning, we have improved the discussion on p.~12 and p.~13 below the list where we discussed the commutation of the resummation with the averaging procedure.

(3) To our knowledge, it is not known whether the Poincaré-Lindstedt method applied in holographic systems affects physical observables. Being inspired by this method, the same is true for our resummation procedure. Considering the energy-momentum tensor obtained before averaging in eq.~(32), it is clear that the resummation has to have some effect on the components since the trace does not vanish and the trace anomaly in curved backgrounds is not satisfied. To restore these results, it is clear that the components of the energy-momentum tensor have to be modified.

To fully answer the question about the effect of the resummation procedure on physical observables, knowledge about the effective theory is required. We briefly touch on this question in our conclusion in the second paragraph of ``Poincar\'e-Lindstedt inspired resummation''. The effect of the PL or our resummation on physical observables is an interesting subject to study, however, due to the lack of an effective theory, we have to postpone studying this in detail to future work.

(4) Originally, the plot was generated without using colours. Therefore, the line was included to mark the border between QNEC being satisfied and violated. In the current version with colours, this line is no longer necessary. We, therefore, follow the recommendation of the referee and removed the line.

(5) It is true that the disorder causes a breaking of the translational symmetry in $\phi$ direction. However, all of our analysis we perform only after disorder averaging the metric. The averaged metric components, displayed in eq.~(25), do not depend on $\phi$, indicating that the translational invariance is restored by the averaging procedure. We therefore think that a mass for the graviton would only be visible before performing the average. To our understanding, the fact that the Einstein vacuum equations are not satisfied is linked to the marginally relevant nature of the disorder.

(6) Indeed, using eq.~(14) it can be obtained straightforwardly to which metric components the disorder contributes. We now include the metric components before expanding in the disorder strength in appendix A, eqs.~66. As can be seen there, the disorder influences $g_{tt}$, $g_{tz}$, $g_{t\phi}$ and $g_{\phi\phi}$. The disorder is such that upon averaging, $\expval{g_{tz}}=0$, i.e., the averaged Poincar\'e patch metric is again in Fefferman--Graham form.

(7) We thank the referee for pointing out these references. We included them in our revised version of the introduction in the first paragraph located entirely on page 3, together with some brief comments on the systems studied therein.

(8) Again, we thank the referee for pointing out these references. We included them in our revised version in the introduction in the first paragraph on page 3 (references [18,19]). When discussing the divergences we find in our analysis, we again comment on these two references on page 9 in the second two last paragraph before section 3 along the following lines. While the non-perturbative analysis explained in these references is very interesting in interpreting the results originally obtained by Hartnoll and Santos (reference [14] in our paper), the setup is different to our approach. We do not consider additional matter fields like the scalar field to introduce disorder, but use the chemical potentials within the metric to source the disorder. Due to this conceptual difference, we do not think that the IR divergences found in our approach can be related to the analysis of 2004.06543 and 2110.11978. It will however be interesting to study how their approach could be adapted to our setup.

(9) Following the suggestions by the referee, we have enlarged the discussion about Anderson localisation in the introduction (second to last paragraph on page 3), also discussing the references pointed out in the report.

Regarding the necessity of quantum effects, we comment the following. Anderson localisation as a phenomenon is not tied to quantumness in the system, but is rather a generic result if waves are considered. While the original paper of Anderson dealt with a tight-binding model where the wave function has the probability interpretation of quantum mechanics, the mathematics behind do not require this interpretation. The notion of localisation can also be found for systems of electromagnetic ([33-35] in the main text) or acoustic waves ([36-38] in the main text) with disorder. This is due to the fact that Anderson localisation results from waves interfering after taking different scattering paths in the disordered potential. Assuming the strong scattering limit, the interference becomes completely destructive, yielding the localisation interpretation. Therefore, the lack of quantum effects does not prohibit Anderson localisation, and correspondingly large $N$ limits may help in shedding light on this phenomenon. We have included this discussion in the introduction in the last paragraph on page 3.

---

## Round 4 · List of Changes

- added the references 1509.02547, 1401.7993, 1505.05171, 1802.08650, 1409.6875, 1511.05970 on prior studies of disorder and commented on their results in the first paragraph located entirely on p. 3 (starting with: ``Einstein-Maxwell theory in D=4 with a disordered ...'')

- enlarged the discussion on Anderson localisation in the introduction in the second to last paragraph on p. 3 (starting with ``The possibility of Anderson localisation in holographic systems ...''). Added and commented on the results regarding Anderson localisation of 1507.00003, 1601.07897, 1602.01067, 1711.10953

- added discussion on the necessity of quantum effects for Anderson localisation in the last paragraph of p. 3, ending on the top of p .4. In this discussion, added the references [33-38]

- added a sentence in the second paragraph on p. 4: ``As we will discuss in more detail in section 2.2, ...''

- added comment on the non-compatibility of the classical equivalence of gravity in the second and first order formulation and the averaging procedure at the end of the paragraph below eq. (20) (p. 7)

- clarified why we expand the exact metric components in \epsilon and \bar\epsilon in the paragraph below eq. (21) (p. 7)

- added a sentence below eq. (24) ``This will be indicated ...''

- added a sentence in the paragraph below eq. (29) ``Since before averaging, the solutions for the charges ...''

- added the references 2004.06543, 2110.11978 and commented on their results on p. 3 in the first paragraph (references [18,19]). Also, added a brief discussion on the relation between their and our approach on p. 9 in the second to last paragraph before section 3

- added a half-sentence below eq. (40) ``..., resulting from the disorder averaging''

- removed the light blue line in figure 1

- enlarged the discussion regarding the commutation of our resummation and the averaging in the paragraph after the list on p. 12

- refined the discussion how to fix the resummation parameters in the paragraph between eqs. (52) and (53) on p. 14

- added an alternative derivation of the resummed metric using the Einstein equations at the bottom of p. 14 and the top of p. 15, in the paragraph containing the new eqs. (57)-(60)

- added a sentence in the second paragraph on p. 16 ``The marginally relevant nature of our disorder ...''

- added a sentence in the third paragraph on p. 16 ``Again, this is due to the marginally ...''

- added exact results for the metric components, before expanding in \epsilon and \bar\epsilon, in Appendix A (eqs. (66a)-(66f))

- added average of the fourth-order expansion of the metric components and the resulting curvature and boundary EM tensor in the new Appendix C

- added numerical evaluation (i.e., without expanding in \epsilon and \bar\epsilon) of the Ricci scalar curvature in the new Appendix D

---

## Round 5 · Referee Report · Anonymous (Referee 5) · 2023-2-13

Report

Dear Authors,

Let me first and mostly address item number 2) of your last reply since it deals with the point that I cannot find convincing about this work.

2)
I would say my main objection to the 'resummation' procedure and therefore to one of the main results of the present work is not yet addressed.
Let me describe the process followed in the manuscript as I understand it, and then state my main objection.
The authors plug in a disordered chemical potential which I understand implies, via the CS gravity formalism, that a disordered source is introduced through the metric (is this correct?). Then they solve the gravity system, first analytically in a perturbative expansion in the disorder strength, and then numerically, for a very realistic disorder (including a lot of modes). Both solutions agree in showing that the IR metric is divergent. This finding is not altogether surprising given that the disorder they introduce is Harris-relevant and thus expected to grow towards the IR.
The authors then take the disordered-average metric, at second order in disorder strength, and look for a somewhat minimalistic modification that will make it finite. They also show that this finite metric solves the vacuum Einstein's equations of motion.
(That is why I asked if the geometry corresponding to the introduction of a source for the metric is generically expected to solve the vacuum Einstein's equations, otherwise I guess one could try a much more generic form for the Einstein's equations).

But, crucially, they cannot show (or I have failed to find it) a direct connection between this solution of the vacuum Einstein's equations and the original problem in the Chern-Simons formalism where the disordered chemical potential is defined. Therefore, to me, the connection between the disordered chemical potential and the smooth solution of the vacuum Einstein's equations that they define as the resummed metric is not proven.

In my view more evidence is needed to establish that the 'resummed' metric describes the correct IR physics corresponding to the introduction of a disordered chemical potential in this system. I would say that Gauge/Gravity tells us that the IR of the dual system is governed by the metric resulting from solving the equations of motion after one has fixed the sources corresponding to the physical problem one wants to study. In this case, after setting a disordered chemical potential, via the Chern-Simons formalism of AdS3 gravity, the authors arrive at an IR-divergent metric, which is not totally surprising (see results like the infinitely disordered IR fixed points). I would expect a very strong evidence would be needed to show that this IR divergent metric is not the physical answer and that instead the IR smooth metric they propose is instead the correct (and of course unique) description of the low energy physics of the dual system.
[An analogy that comes to mind when I consider the situation the authors find when solving the system -an IR divergent metric as a result of switching on some particular sources- is what it is usually done when looking for a dual of a QCD-like theory. There one actually constructs duals where the space ends at a singularity and do not try to 'resumm' it away -since said singularity or end of space is expected to reflect the IR physics of QCD]

As the authors admit, they cannot implement their 'resummation' method in the CS formulation of the problem. But it is in this formulation where it is clear that they are introducing a disordered source via a chemical potential. Thus, the connection between the 'resummed metric' (41) and the original problem seems lacking.
As I said before, and the authors admitted, this is very different to the Poincaré-Lindstedt method they quote, cause in that case it is crystal clear that the resummed metric does result from solving the problem corresponding to the introduction of the original disordered source.

Finally, the authors say that
"By our resummation, we fix all the open resummation parameters uniquely by imposing physically reasonable conditions, or alternatively by also invoking the vacuum Einstein equations. Therefore, the imprint of the original sources of disorder is still present."
I see some problems with this reasoning:
a)As I said, it is not clear to me that a smooth metric has to be found for this problem.
b)Is it clear that the metric corresponding to the disordered problem they introduce via the CS formalism has to solve the vacuum Einstein's equations?
c)Is it the ansatz they take for the resummation the most generic possible and is it obvious why the solution should be obeying a BPS bound (if the 'resummed' metric were to be a solution corresponding to some matter content, should a BPS bound apply at all?)?
d) I think more is needed to prove that 'the imprint of the original sources of disorder is still present'.

3.1) I asked about the energy-momentum tensor averaged over realisations cause indeed it seems that from that point of view, as it should, everything is fine and QNEC conditions and anomaly eqs are fulfilled, right?
Then, one could say that from the point of view of the dual holographic state, as far as the energy-momentum tensor is concerned, there is no need to do a resummation that brings us to a different T_{ij} whose relation to the one corresponding to the original disordered source is not clearly proven.
It might be interesting to compare the T_{ij} the authors report in 3.1) of their reply to the energy momentum tensor after the resummation process.
By the way, is there some metric \gamma missing in the equation for T_{ij}^{ren} in the reply?

---

## Round 5 · Author Response

Dear Editor, we thank the referees for accepting and carefully reading our manuscript as well as for the detailed comments that we address below. A list of all changes to our manuscript is given after our answers to the referees comments.

*** report dated to 29.11.2022

1) We apologise for our somewhat ambiguous formulation, which we clarify in the following. The metric resulting from averaging over realisations in the numerical computation is indeed a function both of $z$ and $\phi$. Of this metric, we compute the Ricci scalar by its definition in terms of the metric and derivatives of the Christoffel symbols, again as a function of $z$ and $\phi$. We state this in the forth sentence of the last paragraph on page 22. However, we do not invoke a numerical package, so both the metric and the resulting Ricci scalar are given by analytical expressions. We chose to do so since this treatment works well at least up to $N=15$, which is sufficient for the arguments we give in the paper. For $N=100$, while the code still works in principle, since we are using analytical expressions, plotting the output for the Ricci scalar becomes extremely lengthy on the hardware available to us. Since the comment about averaging over $N=100$ realisations is not essential to what is discussed in appendix D, we chose to remove the corresponding sentences on the bottom of page 22. We now only state that our code works well at least up to $N=15$, which is sufficient for our analysis.

2) We confirm the observation of the referee: the resummed solution does not agree with the numerical solution. Such a matching is expected in the setup of e.g. ref. [14] of our paper since they are using the Poincaré--Lindstedt method. In our case, while our resummation is inspired by this method, it works fundamentally different. In particular, our resummation is defined such that the singular behaviour in the IR, which as we show by our numerical analysis is clearly present beyond perturbation theory before resummation, is removed. Therefore in the IR, we expect the resummed solution to always be different from the numerical solution. This is the reason why we call our method ``Poincaré--Lindstedt inspired''. In the first paragraph entirely on page 16 we elaborate on this difference. To further indicate this difference, we have added a few words in brackets in page 4 in the beginning of the second to last paragraph, stating that our method is different from the original Poincaré--Lindstedt method of e.g. ref. [14]. We do however emphasise that the relationship between the resummed metric and the geometry induced by the disordered chemical potentials is not arbitrary. By our resummation, we fix all the open resummation parameters uniquely by imposing physically reasonable conditions, or alternatively by also invoking the vacuum Einstein equations. Therefore, the imprint of the original sources of disorder is still present.

2.1) We agree with the referee that finding the resummed metric from solving the vacuum Einstein equations is surprising, at least without using further input. However, the result for the dual energy momentum tensor after resummation (eq. (56) of our paper) is in a form which is compatible with the vacuum Einstein equations. So by knowing this result, we do not expect having to introduce an additional matter field.

3.1) Indeed, computing the energy momentum tensor for each realisation and averaging afterwards yields a healthy result, which in particular also satisfies the trace anomaly equation. We give the corresponding results in the following. Computing the energy momentum tensor for each realisation and averaging results in \begin{align} \langle\langle T_{ij}^{\text{ren}}\rangle\rangle=\frac{c}{288\pi}\begin{pmatrix}-\frac{3}{2}(\bar{\epsilon}-\epsilon)^2 & (\bar{\epsilon}-\epsilon)^2 \ (\bar{\epsilon}-\epsilon)^2 & \frac{1}{2}(\epsilon^2+10\epsilon\bar{\epsilon}+3\bar{\epsilon}^2\end{pmatrix}, \end{align} where $\langle\langle\cdot\rangle\rangle$ denotes the disorder average. Since before averaging, the boundary metric $\gamma^{(0)}$ is still $\phi$-dependent, its Ricci scalar does not vanish. After averaging, it is given by \begin{align} \langle\langle R[\gamma^{(0)}]\rangle\rangle=-\frac{(\epsilon-\bar{\epsilon})^2}{12}. \end{align} Computing the trace of $T_{ij}^{\text{ren}}$ using $\gamma^{(0)}$ and averaging the result yields \begin{align} \langle\langle\tr(T_{ij}^{\text{ren}})\rangle\rangle=-\frac{c(\epsilon-\bar{\epsilon})^2}{288\pi}, \end{align} satisfying \begin{align} \langle\langle\tr(T_{ij}^{\text{ren}})\rangle\rangle=\frac{c}{24\pi}\langle\langle R[\gamma^{(0)}]\rangle\rangle. \end{align} Moreover, $T_{ij}^{\text{ren}}$ is covariantly conserved w.r.t. the boundary metric $\gamma^{(0)}$. In all of these computations, the average was performed at the very end. In particular, the trace of the energy momentum tensor was computed using the energy momentum tensor before averaging. Averaging the determinant of the energy momentum tensor shows that the BPS condition is satisfied. Comparing the above to the energy momentum tensor computed for the averaged metric (eq. (32) in our paper), the trace anomaly equation for curved backgrounds does hold in the above case (eq. (33) in our paper). The value of the trace of the energy momentum tensor is the same in both cases (compare eq. (33) in our paper), but since the boundary metric still depends on $\phi$ in the above approach, the corresponding Ricci scalar does not vanish. Since in our paper, we focus on studying the properties of the averaged geometry, we did not include the above results in the paper.

To the summary: As we discuss in our answer to 2), our resummation, while being inspired by the Poincaré--Lindstedt method of e.g. ref. [14], works fundamentally different. In particular, it renders curvature quantities finite. Therefore, the numerical solution is expected to be different from the resummed solution in the IR. Nevertheless, by our resummation method all resummation parameters are uniquely determined.

---

## Round 5 · List of Changes

- added a comment in brackets on page 4 in the beginning of the second to last paragraph ``(but different from)''
- slightly changed the formulation in the paragraph above eq. (57) on page 15; ``approaches''->``approach'' and ``however''->``remarkably''
- removed the comment about $N=100$ in the last paragraph on page 22, ``This code works well at least up to $N=100$, however computing with the corresponding averaged metric becomes lengthy. In particular, plotting the resulting Ricci scalar unfortunately is restricted by the hardware available to us to $N=15$''.

---

## Round 6 · Referee Report · Anonymous (Referee 1) · 2023-11-1

Report
Let me first apologise for my late reply and thank the authors for their patience and their effort in addressing my questions.
I thank the authors for their clarifications on why the averaged metric (before resummation) cannot be considered as dual to a disordered CFT.
And I also like the new results they add on the implementation of averaging and resummation in the Chern-Simons formalism.
However, the last part of the authors reply brings the focus to a question that still nags me and which I believe needs be clearly addressed in the paper.
The energy momentum tensor (EMT) resulting from averaging the EMT resulting from each disorder realization is healthy (conformal anomaly respected), but very different from that resulting from the resummed metric.
I would then say that this work should clarify the following question:
One can introduce disorder in this Chern-Simons holographic model and (beautifully) solve it obtaining an inhomogeneous metric (for any value of the disorder strength) which results in an EMT that can be computed analytically.
Upon averaging over disorder realizations (this can be nicely done analytically at second order in disorder strength as the authors show or numerically for any disorder) one obtains an EMT that passes the usual checks for a holographic system. Let me call this EMT, EMT_a
However, if one averages the metric over disorder, then the averaged metric is sick, and, unsurprisingly, unphysical issues (violation of QNEC, etc) occur.
Then the authors find a resummation-inspired averaged metric that is smooth and results in a ‘healthy’ energy momentum tensor that I will call EMT_b
What should be the EMT of the dual disordered field theory? EMT_a or EMT_b? And why?
(Related to this, let me ask again a question that might help: have the authors looked at the inhomogeneous metric before averaging? Is it as sick as the resulting averaged metric?)
[This apparent existence of two different answers is, as the authors are aware, quite different from what happens in their ref [14]. There, a physical result like the value of the IR scaling exponent agrees when computed via
a) the resummed metric
b) from averaging the numerical solution that needs not be resummed]
I think that if this question can be discussed and the need for relying in the resummed metric made clear in the paper, this whole saga can be put to rest and the manuscript be finally published in SciPost.
P.S. I could not find any reference in the main text to the results in Appendix D where the behavior of the system beyond perturbative disorder is discussed. (I think it is a nice advantage of this setup the fact that one can quite easily compute the disordered metric at any value of disorder strength without having to solve any differential equation)
I thank the authors for their clarifications on why the averaged metric (before resummation) cannot be considered as dual to a disordered CFT.
And I also like the new results they add on the implementation of averaging and resummation in the Chern-Simons formalism.
However, the last part of the authors reply brings the focus to a question that still nags me and which I believe needs be clearly addressed in the paper.
The energy momentum tensor (EMT) resulting from averaging the EMT resulting from each disorder realization is healthy (conformal anomaly respected), but very different from that resulting from the resummed metric.
I would then say that this work should clarify the following question:
One can introduce disorder in this Chern-Simons holographic model and (beautifully) solve it obtaining an inhomogeneous metric (for any value of the disorder strength) which results in an EMT that can be computed analytically.
Upon averaging over disorder realizations (this can be nicely done analytically at second order in disorder strength as the authors show or numerically for any disorder) one obtains an EMT that passes the usual checks for a holographic system. Let me call this EMT, EMT_a
However, if one averages the metric over disorder, then the averaged metric is sick, and, unsurprisingly, unphysical issues (violation of QNEC, etc) occur.
Then the authors find a resummation-inspired averaged metric that is smooth and results in a ‘healthy’ energy momentum tensor that I will call EMT_b
What should be the EMT of the dual disordered field theory? EMT_a or EMT_b? And why?
(Related to this, let me ask again a question that might help: have the authors looked at the inhomogeneous metric before averaging? Is it as sick as the resulting averaged metric?)
[This apparent existence of two different answers is, as the authors are aware, quite different from what happens in their ref [14]. There, a physical result like the value of the IR scaling exponent agrees when computed via
a) the resummed metric
b) from averaging the numerical solution that needs not be resummed]
I think that if this question can be discussed and the need for relying in the resummed metric made clear in the paper, this whole saga can be put to rest and the manuscript be finally published in SciPost.
P.S. I could not find any reference in the main text to the results in Appendix D where the behavior of the system beyond perturbative disorder is discussed. (I think it is a nice advantage of this setup the fact that one can quite easily compute the disordered metric at any value of disorder strength without having to solve any differential equation)

---

## Round 6 · Author Response

Dear Editor,
first of all we apologise for the delay in our response. We thank the referee for carefully reading our manuscript and for the detailed comments that we address below. A list of all changes of our manuscript is given after our answers to the referees comments.
Below we address the comments of the report dated 13.02.2023.
We confirm the statements by the referee. The disorder is introduced into the system by the chemical potentials of the CS formulation of gravity. By the relation between the CS gauge fields $A$, $\bar{A}$ and the metric, the chemical potentials appear in the metric components, see e.g. app. A, eq. (66). As we commented on p. 8 before starting sec. 2.2, the IR divergent behaviour of the averaged metric is, due to the marginal relevance of the disorder in the sense of the Harris criterion, not unexpected.
The CS gauge fields $A$ and $\bar{A}$, as given eqs. (5-8) with the solutions for ${\cal L}$ and $\bar{\cal L}$ in eqs. (12-13) are solutions to the flatness conditions, eq. (9). This is true for arbitrary $\mu(\phi)$, $\bar{\mu}(\phi)$, so it holds in particular also for our choice of the disorder sources in eqs. (16-17), even after expanding to second order in the disorder strengths. The flatness conditions are equivalent to the vacuum Einstein equations. Since we have a vacuum solution before averaging, we aim to also find vacuum solutions after the average has been implemented.
In the gravity description of QCD, singularities in the IR region are indeed necessary. However, these singular spacetimes still satisfy certain conditions, as discussed e.g. in [1005.4690] (ref.~[41] in our manuscript). As an example, these spacetimes can be used to define the semiclassical approximation for a string. As we discuss around eq. (29), this is not valid for our averaged metric. Moreover, our metric contains closed timelike curves in the deep IR region. Due to these properties, we do not expect that our metric has any ``reasonable'' field theory physics as a dual description. We will further address this below in the answer to point 4 a).
Note also that in the work on disordered IR fixed points, it is the averaged function $\langle A^{(2)}(x,z)\rangle_R$, contained in the metric component $g_{tt}$, which diverges in the IR. However, the Ricci scalar of this metric is finite everywhere. This is not true in our case: the Ricci scalar (eq. (26) in our manuscript) of the averaged metric is divergent in the IR.
a) We will give two arguments for why a modification of our metric is necessary. First, in the naively averaged disordered metric with the IR singularity, the conformal anomaly equation (34) is not fulfilled. Like all quantum anomalies, the conformal anomaly is a UV effect, and switching on an IR relevant (i.e. UV irrelevant) disorder potential should not affect it. On the other hand, our resummed metrics both in the metric and Chern-Simons formulation fulfill the anomaly equation (34), which is a clear-cut criterion for pinning down the free parameters in the resummation. Second, while in AdS/CFT naked singularities can be physically acceptable (c.f. e.g. [hep-th/0002160]), certain admissibility criteria need to be fulfilled. In [1005.4690], it was put forward that a naked singularity is admissible in particular if a string world sheet is repelled by it, i.e. cannot reach the singularity. If it could, the string world sheet fluctuations would grow large, and the string would exit the semiclassical regime. We have shown in equation (29) of the draft that this criterion is violated for the naked singularity in the naively averaged disordered metric. On the other hand, the resummed metrics are completely regular, and this problem ceases to exist. We believe that these two arguments clearly show the unphysical nature of the naked singularity in the naively disordered space-time, which is cured by our resummation.
b) We have already addressed this point above in the second paragraph. The metric defined by the connections of the Chern--Simons formulation, where we introduce the disorder, is a solution to the vacuum Einstein equations. This is true for any choice of $\mu(\phi)$, $\bar{\mu}(\phi)$. This can be understood by the fact that the flatness conditions of the Chern--Simons formalism, which the connections satisfy for any choice of the chemical potentials, are equivalent to the vacuum Einstein equations.
c) In choosing our ansatz for the resummation we have proceeded analogously to the choices made in earlier studies of resummation in disordered systems, such as [1402.0872] and [1504.03288]. These conditions are given in the list in the lower half of p.~12. Subject to these conditions, our ansatz of a general polynomial in $z$ given in eqs. (42-43) is the most generic possible choice.
In our ansatz, we did not allow for non-perturbative dependencies on $z$, such as $\ln z$. Including terms such as $z^n\ln z$ with open coefficients $a_n,b_n$ to the ansatz for the resummation functions $\alpha(z),\beta(z)$ and calculating the Ricci scalar of the resulting metric however shows that these terms do only lead to further divergences. In particular, the open coefficients cannot be fixed such that the divergences already present before resummation get cancelled. Therefore, such terms are excluded.
Regarding the BPS bound, the resummed metric can always be written as a Ba$\tilde{\text{n}}$ados geometry, i.e. a vacuum solution. Therefore, the BPS bound has to hold for the resummed metric.
d) Motivated by the comments of the referee, we took a closer look at implementing the resummation procedure in the CS formulation. In fact, we found a way to perform a resummation in the same spirit as for the metric, in that certain regularity conditions should be satisfied. Due to the disorder, the averaged connections do not satisfy the gauge flatness conditions of the CS theory. By a minimal modification of the connection, in particular of the lowest weight components of the $t$-component, we uniquely determine the modification such that the flatness conditions are satisfied. The metric resulting from the resummed connections does not have curvature singularities; its dual EM tensor does not have a conformal anomaly and the relation between the trace of the boundary EM tensor and the boundary Ricci scalar is satisfied. In particular, up to an overall sign in the $t-\phi$-component, the resulting metric (eq.~(78) in our manuscript) matches what we found by the resummation procedure in the metric formulation (e.g.~eq.~(60) with $c_1=0$ in our manuscript). The sign is due to the fact that averaging and computing the metric from the connections does not commute and can be removed by a time reversal $t\to-t$. These results are discussed in the new section 5 of our manuscript.
Indeed, from the point of view of the energy-momentum tensor averaged only once its components are calculated, nothing particular worrisome appears. Then, no resummation would be necessary. However, as we point out e.g. on page 8 before eq. (25), this is not the approach of our paper. We in particular study the disordered system with the average implemented once the metric components are calculated, following the procedure of [1402.0872] (ref. [14] in our paper). The energy-momentum tensor following from this averaged metric, as we discuss in our paper, does not satisfy the QNEC and the conformal anomaly equation.
Finally, before comparing the $T_{ij}$ of 3.1) of our last reply to the resummed one, we apologise for the typo in the last reply. The r.h.s. of the equation was supposed to display as a matrix with components
\begin{align}
\langle\langle T_{tt}^{\text{ren}}\rangle\rangle&=\frac{c}{288\pi}\left(-\frac{3}{2}(\bar{\epsilon}-\epsilon)^2\right),\\
\langle\langle T_{\phi\phi}^{\text{ren}}\rangle\rangle&=\frac{c}{288\pi}\left(\frac{1}{2}(\epsilon^2+10\epsilon\bar{\epsilon}+3\bar{\epsilon}^2\right),\\
\langle\langle T_{t\phi}^{\text{ren}}\rangle\rangle&=\frac{c}{288\pi}(\bar{\epsilon}-\epsilon)^2=\langle\langle T_{\phi t}^{\text{ren}}\rangle\rangle.
\end{align}
Comparing this result to the resummed energy-momentum tensor, we find that in both cases the trace anomaly equation for curved backgrounds holds, although in different ways. While both $R[\gamma^{(0)}]$ and $\text{tr}(T_{ij}^{\text{ren}})$ vanish after resummation, these quantities are generally non-trivial when calculated before averaging, as shown in our previous reply. There does however not seem to be a deeper relation between the energy-momentum tensor given above and the one obtained after resummation. In particular, the components of the respective energy-momentum tensors behave very differently, without any similarity even for equal disorder strengths. In the resummed energy-momentum tensor, the disorder strengths appear as $\epsilon^2$ and $\bar{\epsilon}^2$ only, but never in a combination $(\epsilon-\bar{\epsilon})^2$ as in the above result. Moreover, in the resummed energy-momentum tensor the $tt$- and the $\phi\phi$-components are equal even for generic disorder strengths, while in the above result, they differ.
With best regards,
M. Dorband, D. Grumiller, R. Meyer and S. Zhao
first of all we apologise for the delay in our response. We thank the referee for carefully reading our manuscript and for the detailed comments that we address below. A list of all changes of our manuscript is given after our answers to the referees comments.
Below we address the comments of the report dated 13.02.2023.
We confirm the statements by the referee. The disorder is introduced into the system by the chemical potentials of the CS formulation of gravity. By the relation between the CS gauge fields $A$, $\bar{A}$ and the metric, the chemical potentials appear in the metric components, see e.g. app. A, eq. (66). As we commented on p. 8 before starting sec. 2.2, the IR divergent behaviour of the averaged metric is, due to the marginal relevance of the disorder in the sense of the Harris criterion, not unexpected.
The CS gauge fields $A$ and $\bar{A}$, as given eqs. (5-8) with the solutions for ${\cal L}$ and $\bar{\cal L}$ in eqs. (12-13) are solutions to the flatness conditions, eq. (9). This is true for arbitrary $\mu(\phi)$, $\bar{\mu}(\phi)$, so it holds in particular also for our choice of the disorder sources in eqs. (16-17), even after expanding to second order in the disorder strengths. The flatness conditions are equivalent to the vacuum Einstein equations. Since we have a vacuum solution before averaging, we aim to also find vacuum solutions after the average has been implemented.
In the gravity description of QCD, singularities in the IR region are indeed necessary. However, these singular spacetimes still satisfy certain conditions, as discussed e.g. in [1005.4690] (ref.~[41] in our manuscript). As an example, these spacetimes can be used to define the semiclassical approximation for a string. As we discuss around eq. (29), this is not valid for our averaged metric. Moreover, our metric contains closed timelike curves in the deep IR region. Due to these properties, we do not expect that our metric has any ``reasonable'' field theory physics as a dual description. We will further address this below in the answer to point 4 a).
Note also that in the work on disordered IR fixed points, it is the averaged function $\langle A^{(2)}(x,z)\rangle_R$, contained in the metric component $g_{tt}$, which diverges in the IR. However, the Ricci scalar of this metric is finite everywhere. This is not true in our case: the Ricci scalar (eq. (26) in our manuscript) of the averaged metric is divergent in the IR.
a) We will give two arguments for why a modification of our metric is necessary. First, in the naively averaged disordered metric with the IR singularity, the conformal anomaly equation (34) is not fulfilled. Like all quantum anomalies, the conformal anomaly is a UV effect, and switching on an IR relevant (i.e. UV irrelevant) disorder potential should not affect it. On the other hand, our resummed metrics both in the metric and Chern-Simons formulation fulfill the anomaly equation (34), which is a clear-cut criterion for pinning down the free parameters in the resummation. Second, while in AdS/CFT naked singularities can be physically acceptable (c.f. e.g. [hep-th/0002160]), certain admissibility criteria need to be fulfilled. In [1005.4690], it was put forward that a naked singularity is admissible in particular if a string world sheet is repelled by it, i.e. cannot reach the singularity. If it could, the string world sheet fluctuations would grow large, and the string would exit the semiclassical regime. We have shown in equation (29) of the draft that this criterion is violated for the naked singularity in the naively averaged disordered metric. On the other hand, the resummed metrics are completely regular, and this problem ceases to exist. We believe that these two arguments clearly show the unphysical nature of the naked singularity in the naively disordered space-time, which is cured by our resummation.
b) We have already addressed this point above in the second paragraph. The metric defined by the connections of the Chern--Simons formulation, where we introduce the disorder, is a solution to the vacuum Einstein equations. This is true for any choice of $\mu(\phi)$, $\bar{\mu}(\phi)$. This can be understood by the fact that the flatness conditions of the Chern--Simons formalism, which the connections satisfy for any choice of the chemical potentials, are equivalent to the vacuum Einstein equations.
c) In choosing our ansatz for the resummation we have proceeded analogously to the choices made in earlier studies of resummation in disordered systems, such as [1402.0872] and [1504.03288]. These conditions are given in the list in the lower half of p.~12. Subject to these conditions, our ansatz of a general polynomial in $z$ given in eqs. (42-43) is the most generic possible choice.
In our ansatz, we did not allow for non-perturbative dependencies on $z$, such as $\ln z$. Including terms such as $z^n\ln z$ with open coefficients $a_n,b_n$ to the ansatz for the resummation functions $\alpha(z),\beta(z)$ and calculating the Ricci scalar of the resulting metric however shows that these terms do only lead to further divergences. In particular, the open coefficients cannot be fixed such that the divergences already present before resummation get cancelled. Therefore, such terms are excluded.
Regarding the BPS bound, the resummed metric can always be written as a Ba$\tilde{\text{n}}$ados geometry, i.e. a vacuum solution. Therefore, the BPS bound has to hold for the resummed metric.
d) Motivated by the comments of the referee, we took a closer look at implementing the resummation procedure in the CS formulation. In fact, we found a way to perform a resummation in the same spirit as for the metric, in that certain regularity conditions should be satisfied. Due to the disorder, the averaged connections do not satisfy the gauge flatness conditions of the CS theory. By a minimal modification of the connection, in particular of the lowest weight components of the $t$-component, we uniquely determine the modification such that the flatness conditions are satisfied. The metric resulting from the resummed connections does not have curvature singularities; its dual EM tensor does not have a conformal anomaly and the relation between the trace of the boundary EM tensor and the boundary Ricci scalar is satisfied. In particular, up to an overall sign in the $t-\phi$-component, the resulting metric (eq.~(78) in our manuscript) matches what we found by the resummation procedure in the metric formulation (e.g.~eq.~(60) with $c_1=0$ in our manuscript). The sign is due to the fact that averaging and computing the metric from the connections does not commute and can be removed by a time reversal $t\to-t$. These results are discussed in the new section 5 of our manuscript.
Indeed, from the point of view of the energy-momentum tensor averaged only once its components are calculated, nothing particular worrisome appears. Then, no resummation would be necessary. However, as we point out e.g. on page 8 before eq. (25), this is not the approach of our paper. We in particular study the disordered system with the average implemented once the metric components are calculated, following the procedure of [1402.0872] (ref. [14] in our paper). The energy-momentum tensor following from this averaged metric, as we discuss in our paper, does not satisfy the QNEC and the conformal anomaly equation.
Finally, before comparing the $T_{ij}$ of 3.1) of our last reply to the resummed one, we apologise for the typo in the last reply. The r.h.s. of the equation was supposed to display as a matrix with components
\begin{align}
\langle\langle T_{tt}^{\text{ren}}\rangle\rangle&=\frac{c}{288\pi}\left(-\frac{3}{2}(\bar{\epsilon}-\epsilon)^2\right),\\
\langle\langle T_{\phi\phi}^{\text{ren}}\rangle\rangle&=\frac{c}{288\pi}\left(\frac{1}{2}(\epsilon^2+10\epsilon\bar{\epsilon}+3\bar{\epsilon}^2\right),\\
\langle\langle T_{t\phi}^{\text{ren}}\rangle\rangle&=\frac{c}{288\pi}(\bar{\epsilon}-\epsilon)^2=\langle\langle T_{\phi t}^{\text{ren}}\rangle\rangle.
\end{align}
Comparing this result to the resummed energy-momentum tensor, we find that in both cases the trace anomaly equation for curved backgrounds holds, although in different ways. While both $R[\gamma^{(0)}]$ and $\text{tr}(T_{ij}^{\text{ren}})$ vanish after resummation, these quantities are generally non-trivial when calculated before averaging, as shown in our previous reply. There does however not seem to be a deeper relation between the energy-momentum tensor given above and the one obtained after resummation. In particular, the components of the respective energy-momentum tensors behave very differently, without any similarity even for equal disorder strengths. In the resummed energy-momentum tensor, the disorder strengths appear as $\epsilon^2$ and $\bar{\epsilon}^2$ only, but never in a combination $(\epsilon-\bar{\epsilon})^2$ as in the above result. Moreover, in the resummed energy-momentum tensor the $tt$- and the $\phi\phi$-components are equal even for generic disorder strengths, while in the above result, they differ.
With best regards,
M. Dorband, D. Grumiller, R. Meyer and S. Zhao

---

## Round 6 · List of Changes

- in the introduction, added the last paragraph on the bottom of page 4, ''Having discussed ...'' until ''... after averaging.'' on the top of page 5.
- in the final paragraph of the introduction, added two sentences referring to sec. 5, ''In section 5 ... in the Chern-Simons formulation.''
- added new section 5 on pages 16-18
- added two paragraphs in the conclusion on page 19, addressing sec .5, ''Apart from the metric formalism ... by a sign after averaging.''
- On the top of page 20, removed the comment about resummation in the Chern-Simons formulation, ''While it is clear how our resum- mation method works on the metric level, it is currently unclear how to implement it in the Chern–Simons formulation. This may require to go beyond the radial gauge (3)-(4).''
All page numbers refer to the resubmitted version of the manuscript

---

## Round 7 · Referee Report · Anonymous (Referee 1) · 2023-12-19

Report

I would like to thank the authors for their effort in clarifying the issue I raised in my last report. I find the results in the new section 5.2 quite interesting and am also happy with the related changes in other parts of the manuscript. (I find it curious that the energy momentum tensor resulting from averaging over realizations vanishes for the case epsilon = \bar epsilon)
I am therefore satisfied with the current version of the article and recommend it for publication in this journal.

---

## Round 7 · Author Response

Dear Editor,
We thank the referee for their additional comments and questions, which we addressed in our last edition of the paper. Below we address the comments of the report dated Nov.01.2023 and the corresponding changes in our manuscript. A list of all changes of our manuscript is given after our answers to the referees comments.

To better highlight the issue addressed by the main question of the referee, we added a statement at the end of our abstract and included a new subsection 5.2 where we discuss this important issue raised by the referee. We summarize the essence of this discussion already in a new paragraph at the end of the introduction (before the organization of the paper), which we quote here:

''Finally, we compare our results with those obtained by first calculating observables for each realisation of the disorder and then averaging only the final result. In particular, we obtain an averaged energy-momentum tensor that differs from the one generated through the Poincaré-Lindstedt-inspired method. The main disadvantage of this, otherwise straightforward, alternative is that there is no way to associate an effective metric with the resulting energy-momentum tensor. In particular, the averaged boundary metric does not provide a source for the averaged boundary energy-momentum tensor.''
Moreover, we also added a brief summary of these statements in the concluding section 6 at the end of p. 20/beginning of p. 21.

We refer now to appendix D in the beginning of section 5 and in the paragraph ''Beyond perturbation theory'' on p. 21.

With best regards,
M. Dorband, D. Grumiller, R. Meyer and S. Zhao

---

## Round 7 · List of Changes

• added half-sentence at the end of the abstract: ''and compare with other approaches to averaging.''
  • added a paragraph on the new subsection 5.2 on the upper half of p. 5: ''Finally, we compare ... energy-momentum tensor.''
  • at the end of the introduction, added one sentence referring to to sec. 5.2: ''In the second part ... the final result.''
  • restructured previous section 5; content of old section 5 is now subsection 5.1, added new subsection 5.2 on p. 18-19
  • added a paragraph in the conclusion at the bottom of p. 20 concerning the new subsection 5.2: ''Furthermore, as discussed ... energy-momentum tensor eq. (79).''
  • added a reference to appendix D in the paragraph ''Beyond perturbation theory'' on p. 21
  • corrected an overall sign typo in appendix A, eq. (85b)
  • remade the plots in fig. 2 to account for the typo (typo did not affect the qualitative statements on the plots, as well as the other results discussed in the manuscript)

---

## Editorial Decision

published